# ONLINE LOW RANK MATRIX COMPLETION

**Prateek Jain**
Google Research
Bangalore, India
`prajain@google.com`

**Soumyabrata Pal**
Google Research
Bangalore, India
`soumyabrata@google.com`

## ABSTRACT

We study the problem of *online* low-rank matrix completion with $M$ users, $N$ items and $T$ rounds. In each round, the algorithm recommends one item per user, for which it gets a (noisy) reward sampled from a low-rank user-item preference matrix. The goal is to design a method with sub-linear regret (in $T$) and nearly optimal dependence on $M$ and $N$. The problem can be easily mapped to the standard multi-armed bandit problem where each item is an *independent* arm, but that leads to poor regret as the correlation between arms and users is not exploited. On the other hand, exploiting the low-rank structure of reward matrix is challenging due to non-convexity of the low-rank manifold. We first demonstrate that the low-rank structure can be exploited using a simple explore-then-commit (ETC) approach that ensures a regret of $O(\text{polylog}(M + N)T^{2/3})$. That is, roughly only $\text{polylog}(M + N)$ item recommendations are required per user to get a non-trivial solution. We then improve our result for the rank-1 setting which in itself is quite challenging and encapsulates some of the key issues. Here, we propose OCTAL (Online Collaborative filTering using iterAtive user cLustering) that guarantees nearly optimal regret of $O(\text{polylog}(M + N)T^{1/2})$. OCTAL is based on a novel technique of clustering users that allows iterative elimination of items and leads to a nearly optimal minimax rate.

## 1 INTRODUCTION

Collaborative filtering based on low-rank matrix completion/factorization techniques are the cornerstone of most modern recommendation systems (Koren, 2008). Such systems model the underlying user-item affinity matrix as a low-rank matrix, use the acquired user-item recommendation data to estimate the low-rank matrix and subsequently, use the matrix estimate to recommend items for each user. Several existing works study this *offline* setting (Candès & Recht, 2009; Deshpande & Montanari, 2012; Jain et al., 2013; Chen et al., 2019; Abbe et al., 2020). However, typical recommendation systems are naturally *online* and interactive – they recommend items to users and need to adapt quickly based on users' feedback. The goal of such systems is to quickly identify each user's preferred set of items, so it is necessary to identify the best items for each user instead of estimating the entire affinity matrix. Moreover, items/users are routinely added to the system, so it should be able to quickly adapt to new items/users by using only a small amount of recommendation feedback.

In this work, we study this problem of the online recommendation system. In particular, we study the online version of low-rank matrix completion with the goal of identifying top few items for each user using say only logarithmic many exploratory recommendation rounds for each user. In each round (out of $T$ rounds) we predict one item (out of $N$ items) for each user (out of $M$ users) and obtain feedback/reward for each of the predictions – e.g. did the users view the recommended movie. The goal is to design a method that has asymptotically similar reward to a method that can pick *best* items for each user. As mentioned earlier, we are specifically interested in the setting where $T \ll N$ i.e. the number of recommendation feedback rounds is much smaller than the total number of items.

Moreover, we assume that the expected reward matrix is low-rank. That is, if $\mathbf{R}_{ij}^{(t)}$ is the reward obtained in the $t^{\text{th}}$ round for predicting item-$j$ for user-$i$, then $\mathbb{E}\mathbf{R}_{ij}^{(t)} = \mathbf{P}_{ij}$, where $\mathbf{P} \in \mathbb{R}^{M \times N}$ is a low-rank matrix. A similar low rank reward matrix setting has been studied in *online* multi-dimensional learning problems (Katariya et al., 2017b; Kveton et al., 2017; Trinh et al., 2020). But in

these problems, the goal is to find the matrix entry with the highest reward. Instead, our goal is to recommend good items to *all* users which is a significantly harder challenge. A trivial approach is to ignore the underlying low-rank structure and solve the problem using standard multi-arm bandit methods. That is, model each *(user, item)* pair as an *arm*. Naturally, that would imply exploration of almost all the items for *each* user, which is also reflected in the regret bound (averaged over users) of $O(\sqrt{NT})$ (Remark 1). That is, as expected, the regret bound is vacuous when the number of recommendation rounds $T \ll N$.

In contrast, most of the existing online learning techniques that leverage structure amongst arms assume a parametric form for the reward function and require the reward function to be convex in the parameters (Shalev-Shwartz et al., 2011, Bubeck, 2011). Thus, due to the non-convexity of the manifold of low-rank matrices, such techniques do not apply in our case. While there are some exciting recent approaches for *non-convex online learning* (Agarwal et al., 2019; Suggala & Netrapalli, 2020), they do not apply to the above mentioned problem.

**Our Techniques and Contributions:** We first present a method based on the explore-then-commit (ETC) approach (Algorithm 2). For the first few rounds, the algorithm runs a pure exploration strategy by sampling random items for each user. We use the data obtained by pure exploration to learn a good estimate of the underlying reward matrix $\mathbf{P}$; this result requires a slight modification of the standard matrix completion result in Chen et al. (2019). We then run the exploitation rounds based on the current estimate. In particular, for the remaining rounds, the algorithm commits to the arm with the highest estimated reward for each user. With the ETC algorithm, we achieve a regret bound of $O(\text{polylog}(M + N)T^{2/3})$ (Thm. 1). This bound is able to get rid of the dependence on $N$, implying non-trivial guarantees even when $T \ll N$. That is, we require only $\text{polylog}(M + N)$ exploratory recommendations per user. However, the dependence of the algorithm on $T$ is sub-optimal. To address this, we study the special but critical case of rank-one reward matrix. The rank-1 setting is itself technically challenging (Katariya et al., 2017b) and encapsulates many key issues. We provide a novel algorithm OCTAL in Algorithm 3 and a modified version in Algorithm 8 that achieves nearly optimal regret bound of $O(\text{polylog}(M + N)T^{1/2})$ (see Theorems 2 and E). The key insight is that in rank-one case, we need to cluster users based on their true latent representation to ensure low regret.

Our method OCTAL consists of multiple phases of exponentially increasing number of rounds. Each phase refines the current estimate of relevant sub-matrices of the reward matrix using standard matrix completion techniques. Using the latest estimate, we jointly refine the cluster of users and the estimate of the best items for users in each cluster. We can show that the regret in each phase decreases very quickly and since the count of phases required to get the correct clustering is small, we obtain our desired regret bounds. Finally, we show that our method achieves a regret guarantee that scales as $\widetilde{O}(T^{1/2})$ with the number of rounds $T$. We also show that the dependence on $T$ is optimal (see Theorem 3). Below we summarize our main contributions ($\widetilde{O}(\cdot)$ hides logarithmic factors):

- We formulate the online low rank matrix completion problem and define the appropriate notion of regret to study. We propose Algorithm 2 based on the explore-then-commit approach that suffers a regret of $\widetilde{O}(T^{2/3})$. (Thm. 1)
- We propose a novel algorithm OCTAL (Online Collaborative filTering using iterAtive user cLustering) and a modified version (Alg. 3 and Alg. 8 respectively) for the special case of rank-1 reward matrices and guarantee a regret of $O(T^{1/2})$. Importantly, our algorithms provide non-trivial regret guarantees in the practical regime of $M \gg T$ and $N \gg T$ i.e when the number of users and items is much larger than the number of recommendation rounds. Moreover, OCTAL does not suffer an issue of large cold-start time (possibly large exploration period) as in the ETC algorithm.
- We conducted detailed empirical study of our proposed algorithms (see Appendix A) on synthetic and multiple real datasets, and demonstrate that our algorithms can achieve significantly lower regret than methods that do not use collaboration between users. Furthermore, we show that for rank-1 case, while it is critical to tune the exploration period in ETC (as a function of rounds and sub-optimality gaps) and is difficult in practice (Lattimore & Szepesvári, 2020)[Ch. 6], OCTAL still suffers lower regret without such side-information (see Figures 1a and 2b).

**Technical Challenges:** For rank-1 case, i.e., $\mathbf{P} = \mathbf{u}\mathbf{v}^\top$, one can cluster users in two bins: ones with $\mathbf{u}_i \geq 0$ and ones with $\mathbf{u}_i < 0$. Cluster-2 users dislike the best items for Cluster-1 users and vice-versa. Thus, we require algorithms that can learn this cluster structure and exploit the fact that within the same cluster, relative ranking of all items remain the same. This is a challenging information to

exploit, and requires learning the latent structure. Note that an algorithm that first attempts to cluster all users and then optimize the regret will suffer the same $\mathsf{T}^{2/3}$ dependence in the worst case as in the ETC algorithm; this is because the difficulty of clustering the users are widely different. Instead, our proposed OCTAL algorithm (Algorithms 3 and 8) in each iteration/phase performs two tasks: i) it tries to eliminate some of the items similar to standard phase elimination method (Lattimore & Szepesvári, 2020)[Ch 6, Ex 6.8] for users that are already clustered, ii) it simultaneously tries to grow the set of clustered users. For partial clustering, we first apply low rank matrix completion guarantees over carefully constructed reward sub-matrices each of which correspond to a cluster of users and a set of active items that have high rewards for all users in the same cluster, and then use the partial reward matrix for eliminating some of the items in each cluster.

## 1.1 RELATED WORKS

To the best of our knowledge, we provide first rigorous online matrix completion algorithms. But, there are several closely related results/techniques in the literature which we briefly survey below.

A very similar setting was considered in Sen et al. (2017) where the authors considered a multi-armed bandit problem with $\mathsf{L}$ contexts and $\mathsf{K}$ arms with context dependent reward distributions. The authors assumed that the $\mathsf{L} \times \mathsf{K}$ reward matrix is low rank and can be factorized into non-negative components which allowed them to use recovery guarantees from non-negative matrix factorization. Moreover, the authors only showed ETC algorithms that resulted in $\mathsf{T}^{2/3}$ regret guarantees. Our techniques can be used to improve upon the existing guarantees in Sen et al. (2017) in two ways 1) Removing the assumption of the low rank components being non-negative as we use matrix completion with entry-wise error guarantees. 2) The dependence on $\mathsf{T}$ can be improved from $\mathsf{T}^{2/3}$ to $\mathsf{T}^{1/2}$ when the reward matrix $\mathbf{P}$ is rank-1.

Multi-dimensional online decision making problems namely stochastic rank-1 matrix bandits was introduced in Katariya et al. (2017b;a); Trinh et al. (2020). In their settings, at each round $t \in [\mathsf{T}]$, the learning agent can choose one row and one column and observes a reward corresponding to an entry of a rank-1 matrix. Here, the regret is defined in terms of the best *(row ,column)* pair which corresponds to the best arm. This setting was extended to the rank $r$ setting (Kveton et al., 2017), rank 1 multi-dimensional tensors (Hao et al., 2020), bilinear bandits (Jun et al., 2019; Huang et al., 2021) and generalized linear bandits (Lu et al., 2021). Although these papers provide tight regret guarantees, they cannot be translated to our problem. This is because, we solve a significantly different problem with an underlying rank-1 reward matrix $\mathbf{P}$ where we need to minimize the regret for all users (rows of $\mathbf{P}$) jointly. Hence, it is essential to find the entries (columns) of $\mathbf{P}$ with large rewards for each user(row) of $\mathbf{P}$; contrast this with the multi-dimensional online learning problem where it is sufficient to infer only the entry ((row,column) pair) in the matrix/tensor with the highest reward. Since the rewards for each user have different gaps, the analysis becomes involved for our OCTAL algorithm. Finally, (Dadkhahi & Negahban, 2018; Zhou et al., 2020) also consider our problem setting but they only provide heuristic algorithms without any theoretical guarantees.

Another closely related line of work is the theoretical model for User-based Collaborative Filtering (CF) studied in Bresler et al. (2014; 2016); Heckel & Ramchandran (2017); Bresler & Karzand (2019); Huleihel et al. (2021). In particular, these papers were the first to motivate and theoretically analyze the collaborative framework with the restriction that the same item cannot be recommended more than once to the same user. Here a significantly stricter cluster structure assumption is made over users where users in same cluster have similar preferences. Such models are restrictive as they provide theoretical guarantees only on a very relaxed notion of regret (termed *pseudo-regret*).

In the past decade, several papers have studied the problem of *offline* low rank matrix completion on its own (Mazumder et al., 2010; Negahban & Wainwright, 2012; Chen et al., 2019; Deshpande & Montanari, 2012; Abbe et al., 2020; Jain et al., 2013; Jain & Kar, 2017) and also in the presence of side information such as social graphs or similarity graphs (Xu et al., 2013; Ahn et al., 2018; 2021; Elmahdy et al., 2020; Jo & Lee, 2021; Zhang et al., 2022). Some of these results namely the ones that provide $\| \cdot \|_\infty$ norm guarantees on the estimated matrix can be adapted into Explore-Then-Commit (ETC) style algorithms (see Sec. 4). Finally, there is significant amount of related theoretical work for online non-convex learning (Suggala & Netrapalli, 2020; Yang et al., 2018; Huang et al., 2020) and empirical work for online Collaborative Filtering (Huang et al., 2020; Lu et al., 2013; Zhang et al., 2015) but they do not study the regret in online matrix completion setting.

## 2 PROBLEM DEFINITION

**Notations:** We write $[m]$ to denote the set $\{1, 2, \ldots, m\}$. For a vector $\mathbf{v} \in \mathbb{R}^m$, $\mathbf{v}_i$ denotes the $i^{\text{th}}$ element; for any set $\mathcal{U} \subseteq [m]$, let $\mathbf{v}_\mathcal{U}$ denote the vector $\mathbf{v}$ restricted to the indices in $\mathcal{U}$. $\mathbf{A}_i$ denotes the $i^{\text{th}}$ row of $\mathbf{A}$ and $\mathbf{A}_{ij}$ denotes the $(i, j)$-th element of matrix $\mathbf{A}$. $[n]$ denotes the set $\{1, 2, \ldots, n\}$. For any set $\mathcal{U} \subset [m], \mathcal{V} \subset [n]$, $\mathbf{A}_{\mathcal{U},\mathcal{V}}$ denotes the matrix $\mathbf{A}$ restricted to the rows in $\mathcal{U}$ and columns in $\mathcal{V}$. Also, let $\|\mathbf{A}\|_{2\to\infty}$ be the maximum $\ell_2$ norm of the rows of $\mathbf{A}$ and $\|\mathbf{A}\|_\infty$ be the absolute value of the largest entry in $\mathbf{A}$. We write $\mathbb{E}X$ to denote the expectation of a random variable $X$.

Consider a system with a set of M users and N items. Let $\mathbf{P} = \mathbf{U}\mathbf{V}^\mathsf{T} \in \mathbb{R}^{\mathsf{M}\times\mathsf{N}}$ be the unknown reward matrix of rank $r < \min(\mathsf{M}, \mathsf{N})$ where $\mathbf{U} \in \mathbb{R}^{\mathsf{M}\times r}$ and $\mathbf{V} \in \mathbb{R}^{\mathsf{N}\times r}$ denote the latent embeddings corresponding to users and items respectively. In other words, we can denote $\mathbf{P}_{ij} \triangleq \langle \mathbf{u}_i, \mathbf{v}_j \rangle$ where $\mathbf{u}_i, \mathbf{v}_j \in \mathbb{R}^r$ denotes the $r$-dimensional embeddings of $i$-th user and the $j$-th item, respectively. Often, we will also use the SVD decomposition of $\mathbf{P} = \bar{\mathbf{U}}\Sigma\bar{\mathbf{V}}$ where $\bar{\mathbf{U}} \in \mathbb{R}^{\mathsf{M}\times r}, \bar{\mathbf{V}} \in \mathbb{R}^{\mathsf{N}\times r}$ are orthonormal matrices i.e. $\bar{\mathbf{U}}^\mathsf{T}\bar{\mathbf{U}} = \mathbf{I}$ and $\bar{\mathbf{V}}^\mathsf{T}\bar{\mathbf{V}} = \mathbf{I}$ and $\Sigma \triangleq \mathsf{diag}(\lambda_1, \lambda_2, \ldots, \lambda_r) \in \mathbb{R}^{r\times r}$ is a diagonal matrix. We will denote the condition number of the matrix $\mathbf{P}$ by $\kappa \triangleq (\max_i \lambda_i)(\min_i \lambda_i)^{-1}$.

Consider a system that recommends one item to every user, in each round $t \in [\mathsf{T}]$. Let, $\mathbf{R}^{(t)}_{u\rho_u(t)}$ be the reward for recommending item $\rho_u(t) \in [\mathsf{N}]$ for user $u$. Also, let:

$$\mathbf{R}^{(t)}_{u\rho_u(t)} = \mathbf{P}_{u\rho_u(t)} + \mathbf{E}^{(t)}_{u\rho_u(t)} \tag{1}$$

where $\mathbf{E}^{(t)}_{u\rho_u(t)}$ denotes the unbiased additive noise. Each element of $\{\mathbf{E}^{(t)}_{u\rho_u(t)}\}_{u\in[\mathsf{M}],t\in[\mathsf{T}]}$ is assumed to be i.i.d. zero mean sub-gaussian random variables with variance proxy $\sigma^2$. That is, $\mathbb{E}[\mathbf{E}^{(t)}_{u\rho_u(t)}] = 0$ and $\mathbb{E}[\exp(s\mathbf{E}^{(t)}_{u\rho_u(t)})] \le \exp(\sigma^2 s^2/2)$ for all $u \in [\mathsf{M}], t \in [\mathsf{T}]$. The goal is to minimize the expected regret where the expectation is over randomness in rewards and the algorithm:

$$\mathsf{Reg}(\mathsf{T}) \triangleq \frac{\mathsf{T}}{\mathsf{M}} \sum_{u\in[\mathsf{M}]} \max_{j\in[\mathsf{N}]} \mathbf{P}_{uj} - \mathbb{E}\big[\sum_{t\in[\mathsf{T}]} \frac{1}{\mathsf{M}} \sum_{u\in[\mathsf{M}]} \mathbf{R}^{(t)}_{u\rho_u(t)}\big]. \tag{2}$$

In this problem, the interesting regime is $(\mathsf{N}, \mathsf{M}) \gg \mathsf{T}$ as is often the case for most practical recommendation systems. Here, treating each user separately will lead to vacuous regret bounds as each item needs to be observed at least once by each user to find the best item for each user. However, low-rank structure of the rewards can help share information about items across users.

**Remark 1.** *If* $\mathsf{T} \gg \mathsf{N}$, *then we can treat each user as a separate multi-armed bandit problem. In that case, in our setting, the well-studied Upper Confidence Bound (UCB) algorithm achieves an expected regret of at most* $O(\sigma\sqrt{\mathsf{NT}\log\mathsf{T}})$ *(Theorem 2.1 in Bubeck & Cesa-Bianchi (2012)).*

## 3 PRELIMINARIES

Let us introduce a different observation model from (1). Consider an unknown rank $r$ matrix $\mathbf{P} \in \mathbb{R}^{\mathsf{M}\times\mathsf{N}}$. For each entry $i \in [\mathsf{M}], j \in [\mathsf{N}]$, we observe:

$$\mathbf{P}_{ij} + \mathbf{E}_{ij} \text{ with probability } p, \quad 0 \text{ with probability } 1 - p, \tag{4}$$

where $\mathbf{E}_{ij}$ are independent zero mean sub-gaussian random variables with variance proxy $\sigma^2 > 0$. We now introduce the following result from Chen et al. (2019):

**Lemma 1** (Theorem 1 in Chen et al. (2019)). *Let rank* $r = O(1)$ *matrix* $\mathbf{P} \in \mathbb{R}^{d\times d}$ *with SVD decomposition* $\mathbf{P} = \bar{\mathbf{U}}\Sigma\bar{\mathbf{V}}^\mathsf{T}$ *satisfy* $\|\bar{\mathbf{U}}\|_{2,\infty} \le \sqrt{\mu r/d}, \|\bar{\mathbf{V}}\|_{2,\infty} \le \sqrt{\mu r/d}$ *and condition number* $\kappa = O(1)$. *Let* $1 \ge p \ge C\mu^2 d^{-1}\log^3 d$ *for some sufficiently large constant* $C > 0$, $\sigma = O\Big(\sqrt{\frac{pd}{\mu^3\log d}}\|\mathbf{P}\|_\infty\Big)$. *Suppose we observe noisy entries of* $\mathbf{P}$ *according to observation model in (4). Then, with probability exceeding* $1 - O(d^{-3})$, *we can compute a matrix* $\widehat{\mathbf{P}}$ *by using Algorithm 4 (Appendix B) with parameters* $(\mathcal{U} = [\mathsf{M}], \mathcal{V} = [\mathsf{N}], \sigma^2, r, p)$ *s.t.,*

$$\|\widehat{\mathbf{P}} - \mathbf{P}\|_\infty \le O\Big(\frac{\sigma}{\min_i \lambda_i} \cdot \sqrt{\frac{\mu d\log d}{p}}\|\mathbf{P}\|_\infty\Big). \tag{5}$$

---

**Algorithm 1** ESTIMATE

---

**Require:** Set of users $\mathcal{U} \subseteq [\mathsf{M}]$, set of items $\mathcal{V} \subseteq [\mathsf{N}]$, total rounds $m$, set of indices $\Omega \subseteq \mathcal{U} \times \mathcal{V}$, rounds in each iteration $b = \max_{u \in \mathcal{U}} |v \in \mathcal{V} \mid (u,v) \in \Omega|$, regularization parameter $\lambda$. Index of round $t$ is relative to the first round when the algorithm is invoked; hence $t = 1, 2, \ldots, m$.

1: **for** $\ell = 1, 2, \ldots, m/b$ **do**
2:     For all $(i,j) \in \Omega$, set $\mathsf{Mask}_{ij} = 0$.
3:     **for** $\ell' = 1, 2, \ldots, b$ **do**
4:         **for** each user $u \in \mathcal{U}$ in round $t = (\ell-1)b + \ell'$ **do**
5:             Recommend an item $\rho_u(t)$ in $\{j \in \mathcal{V} \mid (u,j) \in \Omega, \mathsf{Mask}_{uj} = 0\}$ and set $\mathsf{Mask}_{u\rho_u(t)} = 1$.
                If not possible then recommend any item $\rho_u(t)$ in $\mathcal{V}$ s.t. $(u, \rho_u(t)) \notin \Omega$. Observe $\mathbf{R}^{(t)}_{u\rho_u(t)}$.
6:         **end for**
7:     **end for**
8: **end for**
9: For each $(u,j) \in \Omega$, compute $\mathbf{Z}_{uj}$ to be average of $\lfloor m/b \rfloor$ observations corresponding to user $u$ being recommended item $j$ i.e. $\mathbf{Z}_{uj} = \mathrm{avg}\{\mathbf{R}^{(t)}_{u\rho_u(t)} \text{ for } t \in [m] \mid \rho_u(t) = j\}$. Discard all other observations corresponding to indices not in $\Omega$.
10: Without loss of generality, assume $|\mathcal{U}| \leq |\mathcal{V}|$. For each $i \in \mathcal{V}$, independently set $\delta_i$ to be a value in the set $[\lceil |\mathcal{V}|/|\mathcal{U}| \rceil]$ uniformly at random. Partition indices in $\mathcal{V}$ into $\mathcal{V}^{(1)}, \mathcal{V}^{(2)}, \ldots, \mathcal{V}^{(k)}$ where $k = \lceil |\mathcal{V}|/|\mathcal{U}| \rceil$ and $\mathcal{V}^{(q)} = \{i \in \mathcal{V} \mid \delta_i = q\}$ for each $q \in [k]$. Set $\Omega^{(q)} \leftarrow \Omega \cap (\mathcal{U} \times \mathcal{V}^{(q)})$ for all $q \in [k]$. #*If* $|\mathcal{U}| \geq |\mathcal{V}|$, we partition the indices in $\mathcal{U}$.
11: **for** $q \in [k]$ **do**
12:     Solve convex program

$$\min_{\mathbf{Q}^{(q)} \in \mathbb{R}^{|\mathcal{U}| \times |\mathcal{V}^{(q)}|}} \frac{1}{2} \sum_{(i,j) \in \Omega^{(q)}} (\mathbf{Q}^{(q)}_{i\pi(j)} - \mathbf{Z}_{ij})^2 + \lambda \|\mathbf{Q}^{(q)}\|_\star, \qquad (3)$$

    where $\|\mathbf{Q}^{(q)}\|_\star$ denotes nuclear norm of matrix $\mathbf{Q}^{(q)}$ and $\pi(j)$ is index of $j$ in set $\mathcal{V}^{(q)}$.
13: **end for**
14: Return $\widetilde{\mathbf{Q}} \in \mathbb{R}^{\mathsf{M} \times \mathsf{N}}$ s.t. $\widetilde{\mathbf{Q}}_{\mathcal{U}, \mathcal{V}^{(q)}} = \mathbf{Q}^{(q)}$ for all $q \in [k]$ and for every $(i,j) \notin \mathcal{U} \times \mathcal{V}$, $\widetilde{\mathbf{Q}}_{ij} = 0$.

---

Note there are several difficulties in using Lemma 1 directly in our setting which are discussed below:

**Remark 2** (Matrix Completion for Rectangular Matrices). *Lemma 1 is described for square matrices and a trivial approach to use Lemma 1 for rectangular matrices with* $\mathsf{M}$ *rows and* $\mathsf{N}$ *columns (say* $\mathsf{N} \geq \mathsf{M}$*) by appending* $\mathsf{N} - \mathsf{M}$ *zero rows leads to an undesirable* $(\mathsf{N}/\mathsf{M})^{1/2}$ *factor (Lemma 5) in the error bound (the* $(\mathsf{N}/\mathsf{M})^{1/2}$ *factor does not arise if we care about spectral/Frobenius norm instead of* $\mathsf{L}_\infty$ *norm). One way to resolve the issue is to partition the columns into* $\mathsf{N}/\mathsf{M}$ *groups by assigning each column into one of the groups uniformly at random. Thus, we create* $\mathsf{N}/\mathsf{M}$ *matrices which are almost square and apply Lemma 1 to recover an estimate that is close in* $\mathsf{L}_\infty$ *norm. Thus we can recover an estimate of the entire matrix which is close in* $\mathsf{L}_\infty$ *norm up to the desired accuracy without suffering the undesirable* $(\mathsf{N}/\mathsf{M})^{1/2}$ *factor (Lemma 6 and Steps 10-12 in Algorithm 1).*

**Remark 3** (Observation Models). *The observation model in equation 4 is significantly different from equation 1. In the former, a noisy version of each element of* $\mathbf{P}$ *is observed independently with probability* $p$ *while in the latter, in each round* $t \in [\mathsf{T}]$*, for each user* $u \in [\mathsf{M}]$*, we observe noisy version of a chosen element* $\rho_u(t)$*. Our approach to resolve this discrepancy theoretically is to first sample a set* $\Omega$ *of indices according to equation 4 and subsequently use equation 1 to observe the indices in* $\Omega$ *(see Steps 3-6 in Algorithm 1 and Corollary 1). Of course, this implies obtaining observations corresponding to indices in a super-set of* $\Omega$ *(see Step 5 in Algorithm 1) and only using the observations in* $\Omega$ *for obtaining an estimate of the underlying matrix. In practice, this is not necessary and we can use all the observed indices to obtain the estimate in Step 12 of Algorithm 1.*

**Remark 4** (Repetition and Median Tricks). *The smallest error that is possible to achieve by using Lemma 1 is by substituting* $p = 1$ *and thereby obtaining* $\|\widehat{\mathbf{P}} - \mathbf{P}\|_\infty \leq O\big(\sigma(\min_i \lambda_i)^{-1} \cdot \sqrt{\mu d \log d}\|\mathbf{P}\|_\infty\big)$ *and moreover, the probability of failure is polynomially small in the dimension* $d$*; however, this is insufficient when* $d$ *is not large enough. Two simple tricks allow us to resolve this issue: 1) First we can obtain repeated observations from the same entry of the reward matrix and take its average;* $s$ *repetitions can bring down the noise variance to* $\sigma^2/s$ *2) Second, we can use the*

*median trick where we obtain several independent estimates of the reward matrix and compute the element-wise median to boost the success probability (see proof of Lemma 2).*

We address all these issues (see Appendix B for detailed proofs) and arrive at the following lemma:

**Lemma 2.** *Let rank $r = O(1)$ reward matrix $\mathbf{P} \in \mathbb{R}^{\mathsf{M} \times \mathsf{N}}$ with SVD decomposition $\mathbf{P} = \bar{\mathbf{U}} \mathbf{\Sigma} \bar{\mathbf{V}}^{\mathsf{T}}$ satisfy $\|\bar{\mathbf{U}}\|_{2,\infty} \leq \sqrt{\mu r / \mathsf{M}}, \|\bar{\mathbf{V}}\|_{2,\infty} \leq \sqrt{\mu r / \mathsf{N}}$ and condition number $\kappa = O(1)$. Let $d_1 = \max(\mathsf{M}, \mathsf{N})$, $d_2 = \min(\mathsf{M}, \mathsf{N})$ such that $d_2 = \Omega(\mu r \log(r d_2))$ and $1 \geq p \geq C \mu^2 d_2^{-1} \log^3 d_2$ for sufficiently large constant $C > 0$. Suppose we observe noisy entries of $\mathbf{P}$ according to observation model in (1). For any positive integer $s > 0$ satisfying $\frac{\sigma}{\sqrt{s}} = O\Big(\sqrt{\frac{p d_2}{\mu^3 \log d_2}} \|\mathbf{P}\|_\infty\Big)$, there exists an algorithm $\mathcal{A}$ with parameters $s, p, \sigma$ that uses $m = O\Big(s \log(\mathsf{MN}\delta^{-1})(\mathsf{N}p + \sqrt{\mathsf{N}p \log \mathsf{M}\delta^{-1}})\Big)$ rounds to compute a matrix $\widehat{\mathbf{P}}$ such that with probability exceeding $1 - O(\delta \log(\mathsf{MN}\delta^{-1}))$*

$$\|\mathbf{P} - \widehat{\mathbf{P}}\|_\infty \leq O\Big(\frac{\sigma r}{\sqrt{s d_2}} \sqrt{\frac{\mu^3 \log d_2}{p}}\Big). \tag{6}$$

**Remark 5.** *Alg. $\mathcal{A}$ repeats the following process $O(\log(\mathsf{MN}\delta^{-1}))$ times: 1) sample subset of indices $\Omega \subseteq [\mathsf{M}] \times [\mathsf{N}]$ such that every $(i, j) \in [\mathsf{M}] \times [\mathsf{N}]$ is inserted into $\Omega$ independently with probability $p$. 2) By setting $b = \max_{i \in [\mathsf{M}]} |j \in [\mathsf{N}] \mid (i, j) \in \Omega|$, Algorithm $\mathcal{A}$ invokes Alg. 1 with total rounds $bs$, number of rounds in each iteration $b$, set $\Omega$, set of users $[\mathsf{M}]$, items $[\mathsf{N}]$ and regularization parameter $\lambda = C_\lambda \sigma \sqrt{\min(\mathsf{M}, \mathsf{N})p}$ for a suitable constant $C_\lambda > 0$ in order to compute an estimate of $\mathbf{P}$. The final estimate $\widehat{\mathbf{P}}$ is computed by taking an entry-wise median of each individual estimate obtained as output from several invocations of Alg. 1. Alternatively, Alg. $\mathcal{A}$ is detailed in Alg. 7 in Appendix B.*

Note that the total number of noisy observations made from the matrix $\mathbf{P}$ is $m \cdot \mathsf{M} \geq \mathsf{MN} \cdot p \cdot s$. Therefore, informally speaking, the average number of observations per index is $p \cdot s$ which results in an error of $\widetilde{O}(\sigma / \sqrt{sp})$ ignoring other terms (contrast with error $\widetilde{O}(\sigma / \sqrt{p})$ in equation 5.)

**Remark 6** (Setting parameters $s, p$ in lemma 2)**.** *Lemma 2 has three input parameters namely $s \in \mathbb{Z}, 0 \leq p \leq 1$ and $0 \leq \delta \leq 1$. For any set of input parameters $(\eta, \nu)$, our goal is to set $s, p, \delta$ as functions of known $\sigma, r, \mu, d_2$ such that we can recover $\|\mathbf{P} - \widehat{\mathbf{P}}\|_\infty \leq \eta$ with probability $1 - \nu$ for which the conditions on $\sigma$ and $p$ are satisfied. From (17), we must have $\sqrt{sp} = \frac{c\sigma r}{\sqrt{d_2}} \frac{\sqrt{\mu^3 \log d_2}}{\eta}$ for some appropriate constant $c > 0$. If $r = O(1)$ and $\eta \leq \|\mathbf{P}\|_\infty$, then an appropriate choice of $c$ also satisfies the condition $\frac{\sigma}{\sqrt{s}} = O\Big(\sqrt{\frac{p d_2}{\mu^3 \log d_2}} \|\mathbf{P}\|_\infty\Big)$. More precisely, we are going to set $p = C \mu^2 d_2^{-1} \log^3 d_2$ and $s = \Big\lceil \Big(\frac{c\sigma r \sqrt{\mu}}{\eta \log d_2}\Big)^2 \Big\rceil$ in order to obtain the desired guarantee.*

## 4 EXPLORE-THEN-COMMIT (ETC) ALGORITHM

In this section, we present an Explore-Then-Commit (ETC) based algorithm for online low-rank matrix completion. The algorithm has two disjoint phases of exploration and exploitation. We will first jointly explore the set of items for all users for a certain number of rounds and compute an estimate $\widehat{\mathbf{P}}$ of the reward matrix $\mathbf{P}$. Subsequently, we commit to the estimated best item found for each user and sample the reward of the best item for the remaining rounds in the exploitation phase for that user. Note that the exploration phase involves using a matrix completion estimator in order to estimate the entire reward matrix $\mathbf{P}$ from few observed entries. Our regret guarantees in this framework is derived by carefully balancing exploration phase length and the matrix estimation error (detailed proof provided in Appendix C).

**Theorem 1.** *Consider the rank-$r$ online matrix completion problem with $\mathsf{M}$ users, $\mathsf{N}$ items, $\mathsf{T}$ recommendation rounds. Set $d_2 = \min(\mathsf{M}, \mathsf{N})$. Let $\mathbf{R}_{u\rho_u(t)}^{(t)}$ be the reward in each round, defined as in equation 1. Suppose $d_2 = \Omega(\mu r \log(r d_2))$. Let $\mathbf{P} \in \mathbb{R}^{\mathsf{M} \times \mathsf{N}}$ be the expected reward matrix that satisfies the conditions stated in Lemma 2, and let $\sigma^2$ be the noise variance in rewards. Then, Algorithm 2, applied to the online rank-$r$ matrix completion problem guarantees the following regret:*

$$\mathsf{Reg}(\mathsf{T}) = O\Big(\Big(\mathsf{T}^{\frac{2}{3}}(\sigma^2 r^2 \|\mathbf{P}\|_\infty)^{\frac{1}{3}}\Big(\frac{\mu^3 \mathsf{N} \log d_2}{d_2}\Big)^{1/3} + \frac{\mathsf{N}\mu^2 \|\mathbf{P}\|_\infty}{d_2}\Big) \log^5(\mathsf{MNT}) + \frac{\|\mathbf{P}\|_\infty}{\mathsf{T}^2}\Big). \tag{7}$$

---

**Algorithm 2** ETC ALGORITHM

---

**Require:** users $M$, items $N$, rounds $T$, noise $\sigma^2$, rank $r$ of $\mathbf{P}$, upper bound on magnitude of expected rewards $||\mathbf{P}||_\infty$, no. of estimates $f = O(\log(MNT))$.

1: Set $d_2 = \min(M, N)$ and $v = (N||\mathbf{P}||_\infty)^{-2/3} \left(\frac{T\sigma r}{\sqrt{d_2}} \sqrt{\mu^3 \log d_2}\right)^{2/3}$. Set $p = C\mu^2 d_2^{-1} \log^3 d_2$, $s = \lceil vp^{-1} \rceil$ and $\lambda = C_\lambda \sigma \sqrt{d_2 p}$ for some constants $C, C_\lambda > 0$.
2: **for** $k = 1, 2, \ldots, f$ **do**
3:      For each tuple of indices $(i, j) \in [M] \times [N]$, independently set $\delta_{ij} = 1$ with probability $p$ and $\delta_{ij} = 0$ with probability $1 - p$.
4:      Denote $\Omega = \{(i, j) \in [M] \times [N] \mid \delta_{ij} = 1\}$ and $b = \max_{i \in [M]} |\ |j \in [N] \mid (i, j) \in \Omega|$ to be the maximum number of index tuples in a particular row. Set total number of rounds to be $bs$.
5:      Compute the $k^{\text{th}}$ estimate $\widehat{\mathbf{P}}^{(k)} = \text{ESTIMATE}([M], [N], bs, \Omega, b, \lambda)$. # *(Algorithm 1 is used to recommend items to every user for $bs$ rounds.)*
6: **end for**
7: Compute final estimate estimate $\widehat{\mathbf{P}}$ by taking the entry-wise median of $\widehat{\mathbf{P}}^{(1)}, \widehat{\mathbf{P}}^{(2)}, \ldots, \widehat{\mathbf{P}}^{(f)}$.
8: **for** each of remaining rounds **do**
9:      Recommend $\text{argmax}_{j \in [N]} \widehat{\mathbf{P}}_{ij}$ for each user $i \in [M]$. # *Number of remaining rounds is $T - bsf$.*
10: **end for**

---

**Remark 7** (Non-trivial regret bounds). *Theorem 1 provides non-trivial regret guarantees in the key regime when $N \gg T$ and $M > N$ where the regret scales only logarithmically on $M, N$. This is intuitively satisfying since in each round we are obtaining $M$ observations, so more users translate to more information which in-turn allows better understanding of the underlying reward matrix. However, the dependence of regret on $T$ (namely $T^{2/3}$) is sub-optimal. In the subsequent section, we provide a novel algorithm to obtain regret guarantees with $T^{1/2}$ for rank-1 $\mathbf{P}$.*

**Remark 8** (Gap dependent bounds). *Define the minimum gap to be $\Delta = \min_{u \in [M]} |\mathbf{P}_{u\pi_u(1)} - \mathbf{P}_{u\pi_u(2)}|$ where $\pi_u(1), \pi_u(2)$ corresponds to the items with the highest and second highest reward for user $u$ respectively. If the quantity $\Delta$ is known then it is possible to design ETC algorithms where length of the exploration phase is tuned accordingly in order to obtain regret bounds that scale logarithmically with the number of rounds $T$.*

## 5 OCTAL ALGORITHM

In this section we present our algorithm OCTAL (Algorithm 3) for online matrix completion where the reward matrix $\mathbf{P}$ is rank 1. The set of users is described by a latent vector $\mathbf{u} \in \mathbb{R}^M$ and the set of items is described by a latent vector $\mathbf{v} \in \mathbb{R}^N$. Thus $\mathbf{P} = \mathbf{u}\mathbf{v}^\top$ with SVD decomposition $\mathbf{P} = \bar{\lambda}\bar{\mathbf{u}}\bar{\mathbf{v}}^\top$.

**Algorithm Overview:** Our first key observation is that as $\mathbf{P}$ is rank-one, we can partition the set of users into two disjoint clusters $\mathcal{C}_1, \mathcal{C}_2$ where $\mathcal{C}_1 \equiv \{i \in [M] \mid \mathbf{u}_i \geq 0\}$ and $\mathcal{C}_2 \equiv [M] \setminus \mathcal{C}_1$. Clearly, for all users $u \in \mathcal{C}_1$, the item that results in maximum reward is $j_{\max} = \text{argmax}_{t \in [N]} \mathbf{v}_t$. On the other hand, for all users $u \in \mathcal{C}_2$, the item that results in maximum reward is $j_{\min} = \text{argmin}_{t \in [N]} \mathbf{v}_t$. Thus, if we can identify $\mathcal{C}_1, \mathcal{C}_2$ and estimate items with high reward (identical for users in the same cluster) using few recommendations per user, we can ensure low regret.

But, initially $\mathcal{C}_1, \mathcal{C}_2$ are unknown, so all users are *unlabelled* i.e., their cluster is unknown. In each phase (the outer loop indexed by $\ell$), Algorithm 3 tries to label at least a few unlabelled users correctly. This is achieved by progressively refining estimate $\widetilde{\mathbf{Q}}$ of the reward matrix $\mathbf{P}$ restricted to the unlabelled users and all items (Step 12). Subsequently, unlabelled users for which the difference in maximum and minimum reward (inferred from estimated reward matrix) is large are labelled (Step 19). At the same time, in Step 13 users labelled in previous phases are partitioned into two clusters (denoted by $\mathcal{M}^{(\ell,1)}$ and $\mathcal{M}^{(\ell,2)}$) and for each of them, the algorithm refines an estimate of two distinct sub-matrices of the reward matrix $\mathbf{P}$ by recommending items only from a refined set ($\mathcal{N}^{(\ell,1)}$ and $\mathcal{N}^{(\ell,2)}$ respectively) containing the best item ($j_{\max}$ or $j_{\min}$). We also identify a small set of *good* items for each labelled user (including users labelled in previous phases), which correspond to large estimated rewards. We partition all these users into two clusters ($\mathcal{M}^{(\ell+1,1)}$ and $\mathcal{M}^{(\ell+1,2)}$) such that the set of *good* items for users in different clusters are disjoint. We can prove that such a partitioning is possible; users in same cluster have same sign of user embedding.

---

**Algorithm 3** OCTAL (ONLINE COLLABORATIVE FILTERING USING ITERATIVE USER CLUSTERING)

---

**Require:** Number of users M, items N, rounds T, noise $\sigma^2$, bound on the entry-wise magnitude of expected rewards $||\mathbf{P}||_\infty$, incoherence $\mu$.

1: Set $\mathcal{M}^{(1,1)} = \mathcal{M}^{(1,2)} = \phi$ and $\mathcal{B}^{(1)} = [\mathsf{M}]$. Set $\mathcal{N}^{(1,1)} = \mathcal{N}^{(1,2)} = \phi$. Set $f = O(\log(\mathsf{MNT}))$ and suitable constants $a, c, C, C', C_\lambda > 0$.

2: **for** $\ell = 1, 2, \ldots,$ **do**

3:   Set $\Delta_\ell = C' 2^{-\ell} \min\left(||\mathbf{P}||_\infty, \frac{\sigma\sqrt{\mu}}{\log \mathsf{N}}\right)$.

4:   **for** $k = 1, 2, \ldots, f$ **do**

5:     **for** each pair of non-null sets $(\mathcal{B}^{(\ell)}, \mathsf{N}), (\mathcal{M}^{(\ell,1)}, \mathcal{N}^{(\ell,1)}), (\mathcal{M}^{(\ell,2)}, \mathcal{N}^{(\ell,2)}) \subseteq [\mathsf{M}] \times [\mathsf{N}]$ **do**

6:       Denote $(\mathcal{T}^{(1)}, \mathcal{T}^{(2)})$ to be the considered pair of sets and $i \in \{0, 1, 2\}$ to be its index.

7:       Set $d_{2,i} = \min(|\mathcal{T}^{(1)}|, |\mathcal{T}^{(2)}|)$. Set $p_{\ell,i} = C\mu^2 d_{2,i}^{-1} \log^3 d_{2,i}$ and $s_{\ell,i} = \left\lceil \left(\frac{c\sigma\sqrt{\mu}}{\Delta_\ell \log d_{2,i}}\right)^2 \right\rceil$.

8:       For each tuple of indices $(u, v) \in \mathcal{T}^{(1)} \times \mathcal{T}^{(2)}$, independently set $\delta_{uv} = 1$ with probability $p_{\ell,i}$ and $\delta_{uv} = 0$ with probability $1 - p_{\ell,i}$.

9:       Denote $\Omega^{(i)} = \{(u, v) \in \mathcal{T}^{(1)} \times \mathcal{T}^{(2)} \mid \delta_{uv} = 1\}$ and $b_{\ell,i} = \max_{u \in \mathcal{U}} |v \in \mathcal{V} \mid (u, v) \in \Omega|$. Set total number of rounds to be $m_{\ell,i} = b_{\ell,i} s_{\ell,i}$.

10:    **end for**

11:    Set $m_\ell = \max_{i \in \{0,1,2\}} m_{\ell,i}$.

12:    Compute $\widetilde{\mathbf{Q}}^{(\ell,k)} = \text{ESTIMATE}(|\mathcal{B}^{(\ell)}|, [\mathsf{N}], m_\ell, \Omega^{(0)}, b_{\ell,0}, \lambda = C_\lambda \sigma \sqrt{d_{2,0} p_\ell})$. *# Algorithm 1 is used to recommend items to every user in $\mathcal{B}^{(\ell)}$ for $m_\ell$ rounds.*

13:    For $i \in \{1, 2\}$, compute $\widetilde{\mathbf{P}}^{(\ell,i,f)} = \text{ESTIMATE}(|\mathcal{M}^{(\ell,i)}|, |\mathcal{N}^{(\ell,i)}|, m_\ell, \Omega^{(i)}, b_{\ell,i}, \lambda = C_\lambda \sigma \sqrt{d_{2,i} p_\ell})$. *# Algorithm 1 recommends items to every user in $\mathcal{M}^{(\ell,i)}$ for $m_\ell$ rounds.*

14:  **end for**

15:  Compute $\widetilde{\mathbf{Q}}^{(\ell)} = $Entrywise Median$(\{\widetilde{\mathbf{Q}}^{(\ell,k)}\}_{k=1}^f)$, $\widetilde{\mathbf{P}}^{(\ell,i)} = $Entrywise Median$(\{\widetilde{\mathbf{P}}^{(\ell,i,k)}\}_{k=1}^f)$ for $i \in \{1, 2\}$.

16:  Set $\mathcal{B}^{(\ell+1)} \equiv \left\{u \in \mathcal{B}^{(\ell)} \mid \left|\max_{t \in [\mathsf{N}]} \widetilde{\mathbf{Q}}_{ut}^{(\ell)} - \min_{t \in [\mathsf{N}]} \widetilde{\mathbf{Q}}_{ut}^{(\ell)}\right| \le 2a\Delta_\ell\right\}$

17:  Compute $\mathcal{T}_u^{(\ell+1)} = \{j \in [\mathsf{N}]\} \mid \widetilde{\mathbf{Q}}_{uj}^{(\ell)} + \Delta_\ell > \max_{t \in [\mathsf{N}]} \widetilde{\mathbf{Q}}_{ut}^{(\ell)}\}$ for all $u \in \mathcal{B}^{(\ell)} \setminus \mathcal{B}^{(\ell+1)}$.

18:  For $i \in \{1, 2\}$, for all users $u \in \mathcal{M}^{(\ell,i)}$, compute $\mathcal{T}_u^{(\ell+1)} = \{j \in \mathcal{N}^{(\ell,i)} \mid \widetilde{\mathbf{P}}_{uj}^{(\ell,i)} + \Delta_\ell > \max_{t \in \mathcal{N}^{(\ell,i)}} \widetilde{\mathbf{P}}_{ut}^{(\ell,i)}\}$.

19:  Set $v$ to be any user in $[\mathsf{M}] \setminus \mathcal{B}^{(\ell+1)}$. Set $\mathcal{M}^{(\ell+1,1)} = \{u \in [\mathsf{M}] \setminus \mathcal{B}^{(\ell+1)} \mid \mathcal{T}_u^{(\ell+1)} \cap \mathcal{T}_v^{(\ell+1)} \ne \phi\}$. Set $\mathcal{M}^{(\ell+1,2)} = [\mathsf{M}] \setminus (\mathcal{B}^{(\ell+1)} \cup \mathcal{M}^{(\ell+1,1)})$.

20:  Compute $\mathcal{N}^{(\ell+1,1)} = \bigcap_{u \in \mathcal{M}^{(\ell+1,1)}} \mathcal{T}_u^{(\ell+1)}$, $\mathcal{N}^{(\ell+1,2)} = \bigcap_{u \in \mathcal{M}^{(\ell+1,2)}} \mathcal{T}_u^{(\ell+1)}$.

21:  For $i \in \{1, 2\}$, if $|\mathcal{M}^{(\ell+1,i)}| \le \frac{\mathsf{M}}{\sqrt{\mathsf{T}}}$, then set $\mathcal{B}^{(\ell+1)} \leftarrow \mathcal{B}^{(\ell+1)} \cup \mathcal{M}^{(\ell+1,i)}$ and $\mathcal{M}^{(\ell+1,i)} \leftarrow \phi$.

22: **end for**

---

We also prove that the set of good items contain the best item for each labelled user ($j_{\max}$ or $j_{\min}$). So, after each phase, for each cluster of users, we compute the intersection of *good* items over all users in the cluster. This subset of items (*joint good* items) must contain the best item for that cluster and therefore we can discard the other items (Step 20). We can show that all items in the set of *joint good* items ($\mathcal{N}^{(\ell+1,1)}$ and $\mathcal{N}^{(\ell+1,2)}$) have rewards which is close to the reward of the best item. Therefore the algorithm suffers small regret if for each group of labelled users, the algorithm recommends items from the set of *joint good* items (Step 13) in the next phase. We can further show that for the set of unlabelled users, the difference in rewards between the best item and worst item is small and hence the regret for such users is small, irrespective of the recommended item (Step 12). Note that until the number of labelled users is sufficiently large, we do not consider them separately (Step 21). A crucial part of our analysis is to show that for any subset of users and items considered in Step 5, the number of rounds sufficient to recover a good estimate of the expected reward sub-matrix is small irrespective of the number of considered items (if the number of users is sufficiently large).

**Remark 9** (Practical considerations). *In general OCTAL (Alg. 3) is computationally faster than the ETC Algorithm (Alg. 2) with a higher exploration length. This is because OCTAL eliminates large chunks of items in every phase and therefore has to solve easier optimization problems; on the other hand, ETC has to solve a low rank matrix completion problem in* MN *variables that becomes slower with the exploration length (datapoints). Moreover, OCTAL algorithm runs in phases with the initial*

*phases being very small; hence the users do not have to wait for a long time to even get personalized recommendations like in ETC. These features make OCTAL much more practical than ETC.*

To summarize, in Algorithm 3, the entire set of rounds $[\mathsf{T}]$ is partitioned into phases of exponentially increasing length. In each phase, for the set of unlabelled users, we do pure exploration and recommend random items from the set of all possible items (Step 12). The set of labelled users are partitioned into two clusters; for each, we follow a semi-exploration strategy where we recommend random items from a set of *joint good* items (Steps 13). We now introduce the following definition:

**Definition 3** (($\alpha, \mu$)-Local Incoherence). *For $0 \le \alpha \le 1$, a vector $\mathbf{v} \in \mathbb{R}^m$ is $(\alpha, \mu)$-local incoherent if for all sets $\mathcal{U} \subseteq [m]$ satisfying $|\mathcal{U}| \ge \alpha m$, we have $\|\mathbf{v}_{\mathcal{U}}\|_\infty \le \sqrt{\frac{\mu}{|\mathcal{U}|}} \|\mathbf{v}_{\mathcal{U}}\|_2$.*

Local incoherence for a vector $\mathbf{v}$ implies that any sub-vector of $\mathbf{v}$ having a significant size must be incoherent as well. Note that the local incoherence condition is trivially satisfied if the magnitude of each vector entry is bounded from below. We are now ready to state our main result:

**Theorem 2.** *Consider the rank-1 online matrix completion problem with $\mathsf{T}$ rounds, $\mathsf{M}$ users s.t. $\mathsf{M} \ge \sqrt{\mathsf{T}}$ and $\mathsf{N}$ items. Denote $d_2 = \min(\mathsf{M}, \mathsf{N})$. Let $\mathbf{R}_{u\rho_u(t)}^{(t)}$ be the reward in each round, defined as in equation 1. Let $\sigma^2$ be the noise variance in rewards and let $\mathbf{P} \in \mathbb{R}^{\mathsf{M} \times \mathsf{N}}$ be the expected reward matrix with SVD decomposition $\mathbf{P} = \lambda \bar{\mathbf{u}} \bar{\mathbf{v}}^\mathsf{T}$ such that $\bar{\mathbf{u}}$ is $(\mathsf{T}^{-1/2}, \mu)$-locally incoherent, $\|\bar{\mathbf{v}}\|_\infty \le \sqrt{\mu/\mathsf{N}}$, $d_2 = \Omega(\mu \log d_2)$ and $|\bar{\mathbf{v}}_{j_{\min}}| = \Theta(|\bar{\mathbf{v}}_{j_{\max}}|)$. Then, by suitably choosing parameters $\{\Delta_\ell\}_\ell$, positive integers $\{s_{(\ell,0)}, s_{(\ell,1)}, s_{(\ell,2)}\}_\ell$ and $1 \ge \{p_{(\ell,0)}, p_{(\ell,1)}, p_{(\ell,2)}\}_\ell \ge 0$ as described in Algorithm 3, we can ensure a regret guarantee of $\mathsf{Reg}(\mathsf{T}) = O(\sqrt{\mathsf{T}}\|\mathbf{P}\|_\infty + \mathsf{J}\sqrt{\mathsf{T}\mathsf{V}})$ where $\mathsf{J} = O\Big(\log\Big(\frac{1}{\sqrt{\mathsf{V}\mathsf{T}^{-1}}} \min\Big(\|\mathbf{P}\|_\infty, \frac{\sigma\sqrt{\mu}}{\log \mathsf{N}}\Big)\Big)\Big)$ and $\mathsf{V} = \Big(\max(1, \frac{\mathsf{N}\sqrt{\mathsf{T}}}{\mathsf{M}})\sigma^2 \mu^3 \log^2(\mathsf{MNT})\Big)$.*

Similar to Algorithm 2, Algorithm 3 allows non-trivial regret guarantees even when $\mathsf{N} \gg \mathsf{T}$ provided the number of users is significantly large as well i.e. $\mathsf{M} = \widetilde{\Omega}(\mathsf{N}\sqrt{\mathsf{T}})$.

**Remark 10.** *Under slightly stronger local incoherence conditions on the vector $\bar{\mathbf{u}}$, we can analyze a modified version of* OCTAL *(Alg. 8) without requiring users $\mathsf{M}$ to be large. When $|\mathcal{C}_1| \approx |\mathcal{C}|_2$, the regret guarantee (Thm. E) scales as $\widetilde{O}(\sqrt{\mathsf{NT}/\mathsf{M}})$. Due to space limitations, details of Algorithm 8 and the proof of Theorem E can be found in Appendix E.*

Finally, we show that the above dependence on $\mathsf{N}, \mathsf{M}, \mathsf{T}$ matches the lower bound that we obtain by reduction to the well-known multi-armed bandit problem.

**Theorem 3.** *Let $\mathbf{P} \in [0, 1]^{\mathsf{M} \times \mathsf{N}}$ be a rank 1 reward matrix and the noise variance $\sigma^2 = 1$. In that case, any algorithm for online matrix completion problem will suffer regret of $\Omega(\sqrt{\mathsf{NTM}^{-1}})$.*

# 6 CONCLUSIONS

We studied the problem of online rank-one matrix completion in the setting of repeated item recommendations and blocked item recommendations, which should be applicable for several practical recommendation systems. We analyzed an explore-then-commit (ETC) style method which is able to get the regret averaged over users to be nearly independent of number of items. That is, per user, we require only logarithmic many item recommendations to get non-trivial regret bound. But, the dependence on the number of rounds $\mathsf{T}$ is sub-optimal. We further improved this dependence by proposing OCTAL that carefully combines exploration, exploitation and clustering for different users/items. Our methods iteratively refines estimate of the underlying reward matrix, while also identifying users which can be recommended certain items confidently. Our algorithms and proof techniques are significantly different than existing bandit learning literature. We believe that our work only scratches the surface of an important problem domain with several open problems. For example, Algorithm 3 requires rank-1 reward matrix. Generalizing the result to rank-$r$ reward matrices would be interesting. Furthermore, relaxing assumptions on the reward matrix like stochastic noise or additive noise should be relevant for several important settings. Finally, collaborative filtering can feed users related items, hence might exacerbate their biases. Our method might actually help mitigate the bias due to explicit exploration, but further investigation into such challenges is important.

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
