# OpenReview forum: "Online Low Rank Matrix Completion"
_ICLR.cc/2023/Conference — ICLR 2023 poster_

### Official Review · Reviewer_qsQL · 2022-10-23

**Confidence:** 4
**Correctness:** 3
**Technical Novelty And Significance:** 2
**Empirical Novelty And Significance:** Not applicable
**Recommendation:** 6

**Clarity, Quality, Novelty And Reproducibility:**

Clarity: OK. Easy to follow

Quality and Novelty: In spite that there are some new results, the overall technical contribution beyond existing literature is limited.

Reproducibility: Unknown. Code was not provided.


**Strength And Weaknesses:**

Strength:

1. In spite of a theoretical paper, it is generally easy to follow.

2. The considered online low-rank matrix completion problem is interesting and has received increasing attentions recently.


Weaknesses:

1. Technical novelty & related work: My major concern is its technical contribution beyond existing literature. There have been a few recent developments in contextual low-rank matrix/tensor bandit. The considered online low-rank matrix completion can be viewed as a contextual low-rank matrix bandit setting where the user side is the context and the item side is the action/arm. However, none of the following recent works was compared in the experiments. In my personal opinion, the technical novelty beyond these references are limited.

Sen et al. (2017) considered the same contextual low-rank matrix bandit problem. The authors claim that the proposed work is better as the rate of $T$ is improved from $T^{2/3}$ in Sen et al. (2017) to $T^{1/2}$. However, this improvement is only for rank-1 case. The proposed OCTAL only works for rank-1 and the proposed ETC algorithm has the same $T^{2/3}$ as in Sen et al. (2017).

Lu et al. (2018) used ensemble sampling for the low-rank matrix bandit problem and their framework can be easily adapted to contextual matrix bandit setting. This paper was not mentioned.

Jun et al. (2019) and Huang et al. (2021) considered bilinear bandits that can also be viewed as contextual low-rank matrix bandits. However, the authors just briefly mentioned these references and claimed that "they cannot be translated to our problem".

Zhou et al. (2020) even considered more general low-rank tensor bandit and contextual low-rank tensor bandit settings. In their contextual low-rank tensor bandit settings, some mode of the tensor is context and is given, while other modes of the tensor are arms to be decided. It is a higher-order generalization of the considered online low-rank matrix completion in this paper. Importantly, Zhou et al. (2020) also used ETC-type algorithm and proved $T^{2/3}$, and proposed ensemble sampling-type algorithm for contextual low-rank tensor bandit. Their algorithm can be directly used for the considered problem in this paper.

Lu, X., Wen, Z., and Kveton, B. (2018), “Efficient online recommendation via low-rank ensemble sampling,” Proceedings of the 12th ACM Conference on Recommender Systems, 460–464.

Kwang-Sung Jun, Rebecca Willett, Stephen Wright, and Robert Nowak. Bilinear bandits with low-rank structure. In International Conference on Machine Learning, pp. 3163–3172. PMLR, 2019.

Baihe Huang, Kaixuan Huang, Sham Kakade, Jason D Lee, Qi Lei, Runzhe Wang, and Jiaqi Yang. Optimal gradient-based algorithms for non-concave bandit optimization. Advances in Neural Information Processing Systems, 34:29101–29115, 2021.

Zhou, J., Hao, B., Wen, Z., Zhang, J. and Sun, W.W., 2020. Stochastic Low-rank Tensor Bandits for Multi-dimensional Online Decision Making. arXiv e-prints, pp.arXiv-2007.


2. Although the proposed OCTAL achieves the optimal rate $T^{1/2}$ in time horizon, the algorithm only works for rank-1 case. This is a very limited scenario. In the literature, Katariya et al. (2017) considered rank-1 matrix bandit because their work was the first one to consider such-type problem. Given that there have been many new developments in general-rank matrix bandit setting, it would be important to extend the current OCTAL algorithm and its theory to general-rank case.


3. In practice, how do you estimate rank in the ETC algorithm? In offline matrix completion, we can use cross-validation to tune the rank. However, it is unclear how to tune the rank in online matrix completion case.


4. Related to Question 3, how do you decide other unknown input parameters, e.g., noise \sigma^2, \|P\|_{\infty}, a, c, C, C', C_{\lambda}, in Algorithm 2 and 3?


5. Compared to OCTAL, the proposed ETC has a worse regret bound and a worse numerical performance. What's the motivation of proposing Algorithm 2?



**Summary Of The Paper:**

This paper considers online low-rank matrix completion problem, where in each round the algorithm recommends one item per user, for which it gets a noisy reward sampled from a low-rank user-item preference matrix. The authors proposed two algorithms, ETC and OCTAL. ETC obtains a sub-optimal $T^{2/3}$ rate but the OCTAL only works for rank-1 case.

**Summary Of The Review:**

This paper considers online low-rank matrix completion problem, which can be viewed as a special contextual low-rank matrix bandit. In spite that there are some new results, the technical contribution beyond existing literature is limited. In addition, there are a few unclear parts in the practical implementations.

---

> ### Author Response · Authors · 2022-11-10
> **Response to Review**
>
> We thank the reviewer for the  review. However, we believe that there might be some serious misunderstandings regarding previous work:
>
> 1) ***Sen et. al. 2017*** Please note that although Sen et. al. (2017) considers a qualitatively similar problem, they assume a ***low non-negative rank (a significantly stricter assumption than low rank)***. Furthermore, in order to apply NMF techniques, Sen et. al. (2017) also needed to make strong separability assumptions (see Theorem 1 in their paper). Under these assumptions, they obtain a $T^{2/3}$ regret. In contrast, our result that obtains $T^{2/3}$ regret assumes a low rank structure only - this requires many technically interesting modifications to Offline Low Rank Matrix Completion guarantees (See Remarks 2,3,4 in the paper). Therefore, our ETC algorithm (that obtains a T^{2/3} regret) improves upon existing guarantees in Sen et. al. (2017) in a significant manner by relaxing several stringent assumptions.
> Furthermore, in this problem, the problem of achieving a $\sqrt{T}$ regret guarantee was open to the best of our understanding. Our algorithm makes progress for the rank-$1$ case as it achieves a $\sqrt{T}$ regret guarantee - this is quite involved technically as well (please see the Technical Challenges paragraph in Page 2 for details)  and required many novel ideas. We believe that our ideas and techniques can be used to make progress on the general low rank problem which is a very challenging problem.
>
> 2) ***Lu et. al. (2018), Jun et al  (2019), Zhou et al (2020), Huang et al (2021)*** - As already discussed in details in the Section 1.1, this line of work solves a significantly different problem than ours. This line of work tries to solve the multi-dimensional online learning problem where noisy observations are made corresponding to a 1) entries of a matrix (Lu et. al. , Katariya et. al. Trinh et. al., Kveton et. al. ) 2) entries of tensors (Hao et. al., Zhou et al) 3) Bilinear bandits -  function of a matrix or tensor depending on some chosen context (Jun et al, Hunag et al). However, please note that the goal in all these papers is to minimize the regret with respect to  1) the largest entry of  matrix 2) the largest entry of  tensor 3)   largest value that the function can take respectively. This is ***significantly different*** from our objective; please note that each row of the reward matrix in our setting corresponds to the expected rewards of a particular user and our goal is to minimize the regret of ***all users***.  To summarize, in our language, in multi-dimensional online learning, at each round, they can choose both the user and the item. However, in our setting, we cannot choose the user; instead we recommend items to all users at each round. Hence the problem we consider needs completely novel ideas/techniques.
>
> However, we will make sure to cite Lu et. al. (2018) and Huang et. al. (2021) (difficult in the revised version due to space limitations).
>
> Other Major Questions:
>
> 1) ***$T^{1/2}$ rate achieved by OCTAL is limited contribution:*** As discussed above, the problem of achieving a $O(\sqrt{T})$ regret was open under the low rank assumption. In the only paper (Sen et al 2017) working on a similar flavor, the authors could get a regret of $O(T^{2/3})$ regret via a greedy algorithm under low non-negative rank and stringent separability assumptions. In that regard, we make progress towards solving a difficult problem.
>
> 2) ***Tuning rank and deciding other input parameters:***  Note that an easy solution is to collect a small amount of data at the beginning and use it to tune the rank and other hyper-parameters, which are all  small constants (so should be easy to tune), via cross-validation by treating it as offline data. Often such offline data is also often available apriori (from other sources) for tuning these hyper-parameters in practical scenarios.
>
> Having said that, we agree with the reviewer that hyper-parameter free online algorithms are desirable in practice. But such algorithms with comparable theoretical guarantees remain non-existent even for standard multi-armed bandit problems and is indeed an exciting nascent research area. However, note that this was not the focus of our paper.
>
> 3) ***the proposed ETC has a worse regret bound and a worse numerical performance compared to OCTAL*** -  ETC is a very natural greedy algorithm for any online learning problem and it usually serves as a good baseline for comparison with more complicated algorithms both theoretically and numerically. Secondly, the ETC algorithm works for any rank $r$ but the OCTAL algorithm is tailored towards reward matrices with a rank-$1$ structure. Finally, the ETC algorithm is also quite involved technically (see Remarks 2,3,4 in the paper), relaxes the assumption of low non-negative rank in Sen et. al. 2017 to low rank and also brings forth some of the main ideas. We believe that this is essential for future works based on this research direction.

---

> > ### Comment · Reviewer_qsQL · 2022-11-17
> > **Major issues remain**
> >
> > Thanks the authors for the clarification. My two major issues remain.
> >
> > 1. Regarding to your comments: "To summarize, in our language, in multi-dimensional online learning, at each round, they can choose both the user and the item. However, in our setting, we cannot choose the user; instead we recommend items to all users at each round. Hence the problem we consider needs completely novel ideas/techniques."
> >
> > In fact, in these existing references, Lu et. al. (2018), Jun et al (2019), Zhou et al (2020), they allow a contextual version where the users are contexts and the item is the arm. For example, in Section 4 of https://arxiv.org/pdf/2007.15788.pdf, they also have a contextual version. In this setting, they do not choose the user and instead only choose the item for each user. I might misunderstand this reference. Can you help clarify?
> >
> > 2. The tuning issue is not fully addressed. How do you tune all these parameters in practice using the offline data? There are many unknown parameters in the algorithm. Can you numerically evaluate the methods you mentioned in the experiments? If the offline data is already sufficient to guarantee accurate tuning parameter selection, why do we need to do ETC? Can we just commit based on the parameters estimated from the offline data?

---

> > > ### Author Response · Authors · 2022-11-18
> > > **Response**
> > >
> > > We thank the reviewer for the response and also for asking some great questions. We have provided detailed answers below:
> > >
> > > 1) First, let us talk about the paper the reviewer pointed out specifically i.e.  https://arxiv.org/pdf/2007.15788.pdf, (Zhou et. al. 2022) and Sec 4 in there in particular.
> > >
> > > The reviewer is completely correct in stating that the in the model described in Sec. 4 of Zhou et. al. 2022, the context (or user) is not chosen. We thank the reviewer for bringing this to our attention since we were unaware of this contribution. Indeed, the model described in Sec 4 in this paper can be cast as a generalization of our problem set up with $d_0=1$ and $d-d_0=1$. We will make sure to cite this paper in the updated version. However, please note that Zhou et. al. does not provide any theoretical regret guarantees for this problem and only provides a heuristic algorithm. In fact, in the words of Zhou et. al. themselves, a theoretical analysis of their algorithm is challenging and is left for future work. So, in that regard as well, we do make significant theoretical progress on a very challenging open problem that was indeed formulated by Zhou et. al. before us. We promise to acknowledge this and provide a detailed comparison in the updated version of the paper (difficult now due to space restrictions.) ***We hope that this convinces the reviewer further of the novelty and importance of our work***.
> > >
> > > Please note that this contextual version was not studied in Lu et. al. and Jun et. al. They just study the same multi-dimensional learning problem (where the user can be chosen) as was formulated in Sec 3. of Zhou et. al.
> > >
> > > 2)   Please note that the rank $r$ and the hyper-parameters $a,c,C,C',C_{\lambda}$ described in Step 1 of Algorithm 3  are 1) positive constants and theoretically speaking, they just need to be large enough i.e. they do not scale with the number of  users, number of items or the number of rounds $\mathsf{T}$ 2)  For all the aforementioned hyper-parameters and for the noise $\sigma^2$ and bounds on model parameters $||\mathbf{P}||_{\infty}$, ***we do not need to estimate them well and just need to compute loose upper bounds in practice for these hyper-parameters (which is easy to do with a small amount of data)***.
> > >
> > >
> > > Theoretically speaking as well, at the beginning, we can use $O(\mathsf{T}^{1/4})=o(\sqrt{\mathsf{T}})$ rounds (say) to evaluate the regret (restricted to these few rounds and can be computed) with different values of hyper-parameters (not too many) and choose the set of hyper-parameters that give the minimum regret. This process will  only add a lower order term to the regret and is therefore inconsequential. However, as we mentioned before, this was not the focus of our paper.
> > >
> > > ***Taking such hyper-parameters as input is a very common practice in bandit algorithms. For example, in the very papers the reviewer has pointed out, note that rank, noise and other similar hyper-parameters are also taken as input in Algorithms 1,4 in Zhou et. al. 2022, Algorithm 1  in Lu et al 2018, Algorithm 1 and ESTR in Jun et al 2019.***
> > >
> > > Now, coming to the second part of the question, this is not the case with the model parameters i.e. the reward matrix $\mathbf{P}$. The unknown entries of the reward matrix or rather the gaps between them can scale with rounds $\mathsf{T}$. Moreover, simply obtaining a loose estimate of the reward matrix is not sufficient at all. ***We need a very precise estimate of the entries of $\mathbf{P}$ otherwise we will accumulate large regret that scales with $\mathsf{T}$ whenever we commit for the rest of the rounds***. Now, the offline data (or the online data accumulated for a small number of rounds at beginning) might not be large enough to obtain such a precise estimate. This is the point of the ETC approach 1) We cannot explore too less because then we will not obtain a precise enough estimate - we will accumulate large regret continuously as soon as we commit. 2) We cannot explore too much to obtain a highly precise estimate since the exploration cost will be too high - there exists a sweet spot which we find out in Theorem 1.

---

> > > > ### Author Response · Authors · 2022-11-19
> > > > **Thanks!**
> > > >
> > > > We thank the reviewer for engaging with us and for the great questions. If the concerns of the reviewer are clarified and the reviewer is convinced of the novelty of our work, then can we respectfully request the reviewer to increase their score from 3?
> > > >
> > > > Thanks
> > > > Authors

---

> > > > ### Comment · Reviewer_qsQL · 2022-11-19
> > > > **update score**
> > > >
> > > > Thanks authors for the additional clarification. This improves the understanding of the contribution in this paper. So the contextual bandit algorithm in Zhou et. al. 2022 can also handle the considered problem in this paper, but the former one lacked theoretical justification. In this case, it would be interesting to compare the state-of-the-art with the proposed algorithm in experiments.
> > > >
> > > > Note that Zhou et. al. 2022 also has an ETC-type algorithm with theoretical guarantee of T^{2/3}. Therefore, it is helpful to discuss the additional challenging in the new ETC-algorithm due to the context.
> > > >
> > > > The authors' response on the tuning helps relieve the concern. In the revision, it is helpful to add some remark to discuss this.
> > > >
> > > > I have updated my score to reflect this.

---

> ### Author Response · Authors · 2022-11-16
> **Approaching deadline for end of Discussion phase 1**
>
> We thank the reviewer again for the helpful and detailed review.
>
> Since the deadline for end of Discussion phase 1 is almost there, we just wanted to ask if the concerns raised by the reviewer have been addressed appropriately and if we can clarify anything else.

---

### Official Review · Reviewer_qZSG · 2022-10-24

**Confidence:** 4
**Correctness:** 1
**Technical Novelty And Significance:** 4
**Empirical Novelty And Significance:** 3
**Recommendation:** 8

**Clarity, Quality, Novelty And Reproducibility:**

-Clarity: Good;
-Quality: Good;
-Novelty: Good;
-Reproducibility: Good.

**Details Of Ethics Concerns:**

There are no ethics concerns.

**Strength And Weaknesses:**

Strengths: This work proposed a novel and interesting online low-rank matrix completion problem closely extracted from the application of recommendation systems. The analysis of the proposed method is very inspiring and solid, which utilized the very recent advanced fine-grained theoretical analysis technique in the noisy matrix completion field.

-Weaknesses: This work lacks sufficient experimental comparison with other methods for the recommendation systems application like one or two more offline algorithms besides the baseline UCB.


**Summary Of The Paper:**

This paper considers the online low-rank matrix completion problem for recommendation systems. An expected regret formulation, as the core analysis goal of the proposed problem, is defined and then minimized via the proposed ETC algorithm and a further OCTAL algorithm for special case of rank-one reward matrices. It provides rigorous regret guarantees for the proposed algorithms and supplementary experiments are conducted to verify its superiority over the baselines.

**Summary Of The Review:**

This work formulates an interesting online low-rank matrix completion problem for recommendation systems, and designs several specific algorithms with theoretical guarantees. This work provides lots of solid analysis and proofs, and its originality and novelty are clear enough.

---

> ### Author Response · Authors · 2022-11-10
> **Response to Review**
>
> We thank the reviewer for the appreciative review. Please note that any algorithm that do not use collaboration among users will suffer an extremely poor regret as demonstrated by the UCB algorithm (order-optimal for simple Multi-Armed Bandit setting). - please see Remark 1 in the paper. Nevertheless, if the reviewer suggests any particular algorithm that they would like to see a comparison with, we can definitely update the experiments section with the corresponding comparisons.

---

### Official Review · Reviewer_cVaQ · 2022-11-01

**Confidence:** 4
**Correctness:** 4
**Technical Novelty And Significance:** 4
**Empirical Novelty And Significance:** 4
**Recommendation:** 8

**Clarity, Quality, Novelty And Reproducibility:**


**Novelty:**

This is excellent, impactful work. I admit the novelty is very slightly less than I originally thought due to the existence of the paper [2] which the authors cite in the second paragraph of the related works, but I think the difference is still substantial. Indeed, the algorithm is different, so are the assumptions (low rank instead of low non negative rank), and there is the whole rank 1 case which is very novel and interesting both practically and intuitively from the idea of the proof.

**Clarity, correctness:**


There was an attempt to fix the serious issues I mentioned in my previous review, and I am grateful about this because this is better than what happens in 90 percent of (even more serious) cases.
However, there are still substantial problems. I am **completely certain** that the **current draft** is **not clear** and goes back and forth between formulations of the algorithm. My answer is based on an attempt to reach the “closest correct version” of the results. If I chose the right one, the authors should fix things as per my suggestions, if I chose the wrong one, the authors should still fix things as per their own interpretation.

**TO AUTHORS:** please submit a clean version of the paper and appendix. I will **raise my score to 8** if it is done properly.


**Proof of Corollary 1**:

 In the corollary statement, it is said that the result applies to “algorithm A that recommends random items to users in each round (according to equation 1)”, there is no mention of the strategy of Algorithm 1 where we first sample according to the Bernouilli sampling of  [1] and then subsample a suitable training set from this. Thus, the result is incorrect in its current form.
The authors, again, are not even explaining what they are doing and why. The main part of the proof of the result shows that with high probability, the initial Bernouilli sampling procedure has the property that no single user has been sampled more than m times. That is the end of the proof, no context.
 In the first version of the paper, the authors incorrectly claimed that since this ensures that there are more samples in each line than for the Bernouilli sampling case, performing the algorithm on this dataset will achieve lower error (this does not follow from [1] and it is unclear to what extent it is true and what extra failure probabilities would need to be added).
I suspect the algorithm referred to in corollary 1 should be the algorithm 1 from the main text, with the second line of “line 5” corrected to say “if not possible then recommend any item $\rho_u(t)$ in $\mathcal{V}$ *OUTSIDE $\Omega$*. Observe…” This is hinted at in the proof of Corollary 1 on lines 11-12.  Note that **Line 9 of the algorithm is wrong** here in this interpretation, as all the samples outside $\Omega$ should be discarded: the meaningful averaging procedure should occur later in algorithm 2, in a way that we still need to determine properly during the rebuttal and revisions, see below.
At the end of Corollary 1 (proof), it should be concluded that at the end of the procedure, assuming $\mathcal{F}_1$ doesn’t hold, the sample which is chosen by algorithm A (after discarding samples outside $\Omega$) will in fact be exactly the sample obtained through Bernoulli sampling.
Note there is also  an extra bracket in the main equation in the proof of Corollary 1.

**Algorithm 2, Lemma 2, Lemma 7 and associated proofs**

There is some confusion about the sampling procedure again and how it samples several entries many times. The proof of the appearance of the factor of $1/\sqrt{s}$ is not nearly clean enough to determine the statement without ambiguity and is basically just hand waving. See page 20 of the supplementary, the paragraph that starts with “in our problem….”:  **It is clearly stated here that each sample is sampled (exactly) $s$ times, which is inconsistent with the sampling procedure in algorithm 2** (in the algorithm as written, $f$ independent Bernouilli sampling steps are performed). As far as I can see, the only way for it to be the case that each entry is sampled exactly s times is if we use the same Bernouilli sample a single time, and then go through Algo 1 a total of s independent times (in which case the final training set consists in s identical copies of the Bernouilli sample. This does work, despite the apparent loss of entropy due to removing some randomness. If one goes another way there are more technicalities to deal with in terms of the coupling and the randomness involved in the number of times each entry was sampled.

**Lemma 6:**

In the proof towards the end of page 18 of the supplementary, the authors appear to introduce a condition on $d_1$ and $d_2$ which is not stated in the Lemma statement. If this is needed, it really needs to be reintroduced as an assumption in all relevant results.
In addition, in the paragraph that starts with “Hence, with probability….” $d_1^{-10}$ should be $d_2^{-3}$. This is significant since $d_2$ is smaller than $d_1$, resulting in a larger failure probability. Fortunately, it is written correctly in the lemma statement.







==============================References===================


[1] Yuxin Chen, Yuejie Chi, Jiangqing Fan, Cong Ma, Yuling Yan. “Noisy matrix completion: understanding statistical guarantees for convex relaxation via convex optimization”.  SIAM J. Optimization, 2020.

[2] Rajat Sen, Karthikeyan Shanmugam, Murat Kocaoglu, Alex Dimakis, Sanjay Shakkottai. “Contextual Bandits with latent confounders: an nmf approach”. Artificial Intelligence and Statistics, PMLR 2017.

[3] Miao Xu, Rong Jin, Zhi-Hua Zhou. Speedup Matrix Completion with Side Information: Application to Multi-Label Learning.

[4] Rahul Mazumder, Trevor Hastie, Robert Tibshirani. Spectral Regularization Algorithms for Learning Large Incomplete Matrices. JMLR 2010.



**Strength And Weaknesses:**





**Strengths**:

This is a fantastically interesting and under-researched topic.
There is a lot of material in the paper and the results are non trivial and impactful
There is some progress since the last time I reviewed this paper (ICML): some attempt at fixing the issues I mentioned has been made. In particular, I appreciate that the authors now included the actual optimization problem involved in the estimation of the matrix  (line 12, page 5). I also appreciate that the authors are now conscious of the issue of different sampling regimes (see remark 3), and have attempted to adapt the algorithms to take this into account as per my suggestion. The problems with non square matrices has been much better explained and seems correct now. The case of a large number of observations leading to the same entries being sampled many times is at least attempted.


**Weaknesses**:

The proofs are still very far from clean. Whilst I now agree that all the issues can be fixed “along the lines” sketched in this paper, it has not been done properly yet. In particular, the supplementary is very vague in how it refers to the algorithms is uses, the proof of corollary 1 (which used to be corollary 2 in the previous draft) is still incorrect and largely forgets about all the issues (which are now mentioned in the main paper).

I am not sure, but I remain somewhat sceptical that the resampling procedure proposed in algorithm 2 is the correct one. It might work, but as far as I am concerned, the closest correct version of the (still incorrect) proof of Lemma 7 would use a different strategy.












**Summary Of The Paper:**

This paper provides some regret bounds for *online* matrix completion: at each round, a recommendation is made for each of the M users and the reward is observed. The reward distribution is unchanged through time (this is therefore a contextual bandit scenario).
In the simplest approach (Sections 3, 4, B and C), the authors provide an ETC (explore then commit) approach which roughly consists in making uniformly random (see details in “weaknesses”) predictions until sufficient data has been collected, and then sticking to the optimal strategy (recommending the “best” item according to the current estimate of the matrix) for the rest of the time. The estimation procedure is based on nuclear norm regularization. A regret bound of $\widetilde{O}(T^{2/3})$ is provided for this setting.

In the next approach, the specific case where the ground truth data is rank one is studied via an ingenious algorithm called OCTAL which leverages the fact that the users and items can be clustered based on the sign of their latent representations. Regret bounds are proved for this setting and experiments demonstrate the efficiency of the method



**Summary Of The Review:**

It is a **very interesting and impactful paper**. The *proofs are still quite sloppy*, though they improved since the last time I reviewed this paper two conferences ago.

Fortunately, ICLR is openreview and there is unlimited space for rebuttals and for updated versions of the manuscript, so we will definitely get to the bottom of this. I will raise my score to 8 if the next revision uploaded contains *clean proofs*.





[**BEGINNING OF POST REBUTTAL UPDATE AND SUMMARY**]


I have previously reviewed this paper. This is a highly technical and complex paper that I am guessing required a lot of work from the authors. The original version had both minor typos and significant technical issues with a lot of vaguely phrased statements in between, around 60 percent of which eventually turned out to be correct and 40 percent of which could be fixed with modifications to the assumptions or calculations.  After discussions and rebuttals, we managed to fix all issues and improve the readability of the proofs and the versions of the manuscript are converging. Therefore I think the current version of the paper is ready for publication and is a very interesting and worthy piece of work, with **substantially above average quality and impact compared to other ICLR accepted papers.** Therefore, I am raising my score from 5 to 8. **I would put a score of 9 if it was possible.**

I have also increased my correctness score from 2 to 4. However, there are some parts of the paper I didn't have time to look at in much detail during this round of review (mainly Appendix D), so I highly encourage the authors to proofread this before the camera-ready version.

In general, I wish authors would write more details in calculations in other submissions. Otherwise, if there are errors (which is almost unavoidable in such highly technical papers), reviewers will end up (legitimately) questioning the correct parts of the papers as well.


[**END OF POST REBUTTAL UPDATE AND SUMMARY**]




==========================Miscellaneous extra comments (**Minor**)============================



You might want to cite [4] when talking of the nuclear norm regularisation strategy as this is the usual reference from a practical point of view.

In the paragraph starting with “In the past decade, …” on page 3, the authors mention “inductive matrix completion”, i.e. the problem of matrix completion in the presence of side information.  It is usual to cite at least [3] there.


Theorem 1 from [1] is correctly cited in page 17 of the supplementary (as “Lemma 4”) but not as “lemma 1” in the main paper, where there is an arbitrary constant c there. I agree with remark 11 that the result can probably be extended this way but that is not the statement of the main theorem (1.3) in that reference as is (though there is an extension in the same paper which is formulated similarly and the “for a constant c” is still lying around in the theorem statement in that reference without being used). In addition, it doesn’t seem that the authors need this strengthened result here  as the most likely place where this could be needed would be the end of the proof of Lemma 6, but there is an independent failure probability there which means it is not necessary.





==============================References===================


[1] Yuxin Chen, Yuejie Chi, Jiangqing Fan, Cong Ma, Yuling Yan. “Noisy matrix completion: understanding statistical guarantees for convex relaxation via convex optimization”.  SIAM J. Optimization, 2020.

[2] Rajat Sen, Karthikeyan Shanmugam, Murat Kocaoglu, Alex Dimakis, Sanjay Shakkottai. “Contextual Bandits with latent confounders: an nmf approach”. Artificial Intelligence and Statistics, PMLR 2017.

[3] Miao Xu, Rong Jin, Zhi-Hua Zhou. Speedup Matrix Completion with Side Information: Application to Multi-Label Learning.

[4] Rahul Mazumder, Trevor Hastie, Robert Tibshirani. Spectral Regularization Algorithms for Learning Large Incomplete Matrices. JMLR 2010.

---

> ### Author Response · Authors · 2022-11-10
> **Response to Review**
>
> First of all, we thank the reviewer profusely for the highly constructive feedback. Indeed, it has been extremely useful. We have updated the paper with a revised version that contains the following (most significant) changes
>
> 1) Algorithm 1 has been updated to reflect the changes as correctly suggested by the reviewer. Remark 5, Lemma 2 and Theorem statements has also been updated to correct minor issues in the previous version (see below for more details).
>
> 2) Appendix B and associated lemmas/corollaries have now been updated and extended significantly to make the statements/ideas  precise and detailed. Every Lemma is now associated with an algorithm box to explain the strategy in details. Before the proofs of Corollary 1, Lemma 6 and Lemma 7 (restatement of Lemma 2) in Appendix B, we have now added  overviews/intuition of technical modifications so that the proofs (and their reasonings) are easy to follow.
>
> Below, we provide answers to the major questions raised by the reviewer.
>
> 1) ***Proof of Corollary 1:***
>
> The reviewer has correctly pointed out that in Algorithm 1, it should be the following "if not possible then recommend $\rho_u(t)$ in $\mathcal{V}$ outside $\Omega$.  Indeed, our aim is to obtain reward observations corresponding to all indices in the sampled Bernoulli mask $\Omega$; all observations outside $\Omega$ that are made in the process are discarded.
> If we only use the observations corresponding to indices in $\Omega$ to reconstruct the low rank matrix, we can directly use the guarantees in Chen et. al. (which is what we do). The updated Algorithm 1 (Line 5 and Line 9 in Algorithm 1) reflects these changes. Please note that in Algorithm 1, we obtain repeated observations ($s=m/b$ times in the For Loop in Line 1 - 8 in Alg. 1)  corresponding to indices in the same fixed Bernoulli Mask $\Omega$  and the ***averaging is done in Step 9 in Alg. 1*** to reduce the noise variance proxy to $\sigma^2/s$. Note that the mask $\Omega$ is an input parameter to Algorithm 1 that is sampled in Algorithms 2,3 which in turn invoke Algorithm 1. Alternatively, please see Algorithms 6, 7, the detailed discussion above Corollary 1 and  Lemma 7 and their proofs which have been updated extensively taking into account the suggestions of the reviewer. We hope that these clarify our strategy.
>
> 2) ***Algorithm 2, Lemma 2, Lemma 7 and associated proofs***
>
> "***It is clearly stated here that each random variable corresponding to a particular tuple of indices is sampled (exactly) $s$  times***"- this is happening in Algorithm 1 itself in the For Loop in Lines 1-8 in Algorithm 1 where $s=m/b$ since the same Bernoulli Mask $\Omega$ is used for repeated sampling and reduction of noise variance proxy to $\sigma^2/s$. We have now made the idea and proof for the variance reduction step detailed in Appendix B. Please refer to Algorithm 7, the detailed discussion above Lemma 7 (see ***Repeated recommendations for small estimation error***) and the proof of lemma 7. Indeed, as pointed out by the reviewer, Algorithm 1 goes through the same Bernoulli mask $s=m/b$ times (For Loop in Step 1 of Algorithm 1).
>
> Also, there might be a slight misunderstanding. Please note that the $f$ independent estimates corresponding to independent (and different) Bernoulli masks sampled  (in Algorithm 2- For Loop in Line 2 and in Algorithm 3 - For Loop in line 4)  are used for increasing the probability of success by a median trick (entry-wise median of the independent estimates- Line 7 in Algorithm 2 and Line 15 in Algorithm 3). ***This step is not used for the variance reduction but for increasing the success probability for computing a good estimate of the low rank matrix/sub-matrix.*** Again, please refer to Appendix B (Algorithm 7, discussion above Lemma 7 (Independent estimates and entry-wise median for small error probability) and the proof of Lemma 7) for details.
>
> 4) ***Lemma 6***
>
> Indeed, we missed the condition on $d_2$. We altered the probability of error in the application of Bernstein's inequality so that the condition required becomes very mild - we only need $d_2=\Omega(\mu r \log (rd_2))$. Since $\mu$,$r$ is usually considered to be constants in matrix completion problems, we only need $d_2$ to be a large constant. Nevertheless, this condition now appears in Lemma 2 and all the main theorem statements.
>
> Minor questions
>
> 1) We have now made the appropriate citations as suggested by the reviewer in the revised version.
> 2) We also changed Lemma 1 as suggested by the reviewer.

---

> ### Author Response · Authors · 2022-11-16
> **Approaching deadline for end of Discussion Phase 1**
>
> We thank the reviewer again for the extremely helpful and detailed review.
>
> Since the deadline for end of Discussion phase 1 is almost there, we just wanted to ask if the questions raised by the reviewer have been addressed appropriately and if we can clarify anything else.

---

> > ### Comment · Reviewer_cVaQ · 2022-11-16
> > **Factors of s**
> >
> > Dear authors,
> >
> > **Apologies for the late reply**, I was originally planning on giving a more detailed answer and didn't realise how close the deadline was. **The corrections to the sampling procedure do seem to make sense now**. I am impressed. Thanks! : )
> >
> >
> > However, note that the factors of s in the statement and proof of lemma 7 still contain at least one error (line 5 of the statement). I know this is minor, but this is still there despite the fact that I pointed this out in my first review.   It makes me worry about what the parts of the paper I didn't have time to look at in greater detail are actually like.
> >
> > Before I raise my score, could you fix both this this **and also any other similar errors I might not have seen**?  Don't worry, as promised, even if the deadline has passed, I will raise my score to 8 if I cannot find any inconsistent notations or problems anymore.
> >
> > I apologize again for the inconvenience and the late reply.
> >
> >
> > Looking forward to hearing back from you and seeing the revised manuscript.
> >
> > **Congratulations again on the nice paper and good luck with the proofreading**,

---

> > > ### Author Response · Authors · 2022-11-16
> > > **Response**
> > >
> > > We thank the reviewer for the response.
> > >
> > > If we understand correctly, the reviewer is pointing to the extra factor of $\sqrt{s}$ in the LHS of the display $\frac{\sigma}{\sqrt{s}}=O\Big(\sqrt{\frac{pd_2}{\mu^3\log d_2}}\|\|\mathbf{P}\|\|_{\infty}\Big)$ in Line 5 of the statement of Lemma 7.
> > >
> > > Please note that this is not an error. As we have mentioned in the proof,  after observing the entries of the sampled Bernoulli Mask $s$ times, we can take an average at each entry and hence reduce the noise variance proxy to $\sigma^2/s$. However, if we now apply the guarantees in Corollary 1 (which points to Lemma 6 in turn for the necessary conditions), then the condition $\sigma=O\Big(\sqrt{\frac{pd_2}{\mu^3\log d_2}}\|\|\mathbf{P}\|\|_{\infty}\Big)$
> > >
> > > becomes $\frac{\sigma}{\sqrt{s}}=O\Big(\sqrt{\frac{pd_2}{\mu^3\log d_2}}\|\|\mathbf{P}\|\|_{\infty}\Big)$ (the noise variance proxy is now $\sigma^2/s$).
> > >
> > > We have also updated the proof of Lemma 7 in the supplementary with additional explanation to make this clear.
> > >
> > > As suggested by the reviewer, we are also taking another detailed pass over the paper and the appendix. We will make sure to submit another version before the deadline if we find any inconsistencies/errors in the paper.

---

> > > > ### Comment · Reviewer_cVaQ · 2022-11-16
> > > > **Thanks!**
> > > >
> > > > Dear Authors,
> > > >
> > > > Many apologies. You are right! I was a bit confused by the fact that the condition has the denominator and numerators reversed compared to the result statement!
> > > >
> > > >
> > > > Best

---

> > > > > ### Author Response · Authors · 2022-11-16
> > > > > **Thanks to you as well**
> > > > >
> > > > > No worries at all. Please do let us know if we can clarify anything else.

---

> > ### Comment · Reviewer_cVaQ · 2022-11-18
> > **Medians**
> >
> > Dear Authors,
> >
> >
> > Apologies again for the late replies. Also apologies in case I misunderstood something. Still, I think further cleaning up the paper will benefit the community greatly.
> >
> >
> > I understand the algorithm better now and the fact that you use the mean first and then the median to increase the probability of success. By the way I also liked the paragraph starting with "for the sake of simplicity" in page 26. I think this was in response to one of my earlier reviews.
> >
> > However,  the theoretical implications are not properly explained. **How does the median affect the final estimates?** It seems that Lemma 7 is not actually using the fact that the final estimate is a median rather than a mean (though I do understand that at least intuitively the median is better if the aim is to increase the probability of success).  Are you actually using this fact?  If you are, what property of the median are you using and could you write it down as a separate lemma statement?
> >
> > **Note that in the proof of Lemma 7, you clearly state "since with probability 9/10, each estimates satisfy the guarantee (18)"  Apart from the use of 9/10 which is introduced and  then translated back into a $1-\delta$ type probability for no reason, it seems there is an issue in the sense that if you really actually require EACH estimate to satisfy the condition, the use of the median doesn't change anything to the proof and any result you do obtain would hold for any arbitrarily chosen value in the convex hull of the f estimates.**
> >
> >
> >
> > **What worries me a bit more is that it does look like you do reduce the probability of failure again after equation (19) but this is definitely not properly justified.**
> >
> >
> > As far as I am concerned I cannot see how the median trick over independent estimates of the matrix actually allows you to reduce the failure probability without at least using another theorem that is unstated. You say you use a Chernov bound again (above equation (19))  but this is a vague statement and please note that you cannot use it component wise.  **Are you using some kind of Chernov bound for medians of matrices?
> > Note that although there is independence between different samples of the same matrix, but there is no independence between the errors for different entries. There is a maximum over entries of the error hanging around so you definitely have to be much more careful than you are being in the current draft. **
> >
> >
> >
> >
> > For the avoidance of doubt, I do, still, think this can be fixed, and the final results hold (at least in O notation). But you might need to remove the median step with f and increase both p and s instead.
> >
> >
> >
> > Remaining typos:
> >
> >  Bracket not closed at the top of page 26
> >
> > Bottom of page 25 "Conditioned on the event that the exploration fails (and therefore the exploration stage as well)..." (the second "exploration" should be "exploitation"
> >
> > In line 4 of the proof of theorem 1 on page 25,  I still feel like "In that case" is a bit confusing. I think you mean something like "Consequently".

---

> > > ### Author Response · Authors · 2022-11-18
> > > **Response to Medians**
> > >
> > > We thank the reviewer for the follow-up question. We believe that there are some slight misunderstandings but please note that all we are doing is a simple and standard median trick - there is no error in the median analysis. We hope that the detailed answer below will clarify (please refer to the revised version of the paper):
> > >
> > >
> > > Note that in Page 24 above eq. (18), we have clearly mentioned and proved the following:
> > >
> > > Fix any pair of indices $(i,j)$. For any of the estimates $\widehat{\mathbf{P}}^{(z)}$ indexed by $z\in [f]$, it satisfies eq. (18) in the paper with probability at least $9/10$.  Our goal is to increase the probability of success to $1-\delta$  and we will show that the entry-wise median of the $f$ estimates does have the increased probability of success.
> > >
> > > Let us be very detailed here: consider the random variable
> > > $Y^{(z)}$ which is an indicator random variable that takes $1$ when eq. (18) in the paper is satisfied for index $(i,j)$ in the estimate $\widehat{\mathbf{P}}^{(z)}$. Clearly $Y^{(z)}$ is $1$ with probability at least $9/10$.
> > > Since the estimates $\widehat{\mathbf{P}}^{(1)}, \dots,\widehat{\mathbf{P}}^{(f)}$ are independently computed , the random variables $Y^{(1)}, Y^{(2)}, \dots, Y^{(f)}$ are independent as well. We compute $\widehat{\mathbf{P}}$ where each entry is the median of the corresponding entries in the $f$ computed estimates.
> > >
> > >  Hence consider the median $\widehat{\mathbf{P}}_{ij}$ of the $ij^{th}$ entry of the $f$ independent estimates. Note that the computed median will satisfy eq. (19) in the paper
> > > if at least half of the random variables $Y^{(1)}, Y^{(2)}, \dots, Y^{(f)}$ are non-zero.  Hence, we can apply Chernoff bound directly to state that
> > >
> > > $\Pr(\sum_{i=1}^{f} Y^{(i)} < f/2 ) \le 2\exp(-4f/75)$
> > >
> > > Therefore, by setting $f =O(\log (\mathsf{MN}\delta^{-1}))$, we must have that  $\Pr(\sum_{i=1}^{f} Y^{(i)} < f/2 ) \le \delta/\mathsf{MN}$
> > >
> > > implying that eq. (19) in the paper holds with probability $1-\delta/\mathsf{MN}$.
> > >
> > > Now, as the reviewer mentions, we do not have independence among the entries. However, we can take a union bound over all the $\mathsf{MN}$ entries which is what we do in going from eq. (19) to eq. (20) in the paper. So by taking a union bound over all pairs of indices (at most $\mathsf{MN}$ of them) we get that for all pairs of indices, we have eq.(20) in the paper
> > > with probability $1-\delta$
> > >
> > > ***Couple of important points to note below:
> > >
> > > 1) Each of the estimates satisfies the condition in (18) with probability at least 9/10, not with probability $1-\delta$.  We prove that the entry-wise median of the estimates satisfy the guarantee (eq. (19) with probability $1-\delta/\mathsf{MN}$. There is no reduction in the failure probability when going from eq. (19) to eq. (20). We are simply taking a union bound over all pairs of indices to show that eq. (20) holds with probability $1-\delta$.
> > >
> > > 2) We do not assume any independence among the entries of an estimate. We only use the fact that for a fixed index, we have independence among different samples of the same matrix. (as the reviewer mentions)
> > >
> > > 3) Increasing $s$ and $p$ will only decrease the estimation error and not increase the success probability.
> > > ***
> > >
> > > Taking the suggestion of the reviewer into account, we have now provided claim environments within Lemma 7 to clarify our statements. We have also made the proof more detailed as suggested by the reviewer.

---

> > > > ### Comment · Reviewer_cVaQ · 2022-11-18
> > > > **Thanks!**
> > > >
> > > > [**MEDIAN ISSUE RESOLVED. 9/10 issue was related**] Thanks a lot. That make sense! Thanks for being so responsive as well.
> > > >
> > > > Sorry about all the bother... but I am sure you understand as a reviewer who has to review 5-6 papers with 20+ pages of supplementary each (in some cases involving typos or errors), having some minimum level of detail in the proofs helps...It could be a matter of opinion but I definitely feel like the extra page of details you added (claim 2) was necessary to make the paper more accessible to readers.
> > > >
> > > >
> > > >
> > > > [Edit: clarified in the title summary for the meta reviewer that the 9/10 issue was also a misunderstanding due to the skipped (but correct) steps in the consequences of the use of the median]

---

> > ### Comment · Reviewer_cVaQ · 2022-11-18
> > **Another important question**
> >
> >
> > Dear Authors,
> >
> > Sorry about all the bother but I do have another pretty important question about the result from Chen et al and your own application of it.
> >
> >
> > For a while I actually thought it could translate to an  error bound with decay $\sigma\sqrt{\frac{ d\log(d)}{n}}$ where $n$ is the expected number of samples (i.e. $pd^2$). It is true that the *conditions* on $\sigma$ and $p$ (which are required for the error bound to hold in the first place) are roughly of this order. However, looking at the actual error bound itself, it now seems to me that it scales like $\sigma\sqrt{\frac{ d^3\log(d)}{n}}$. Indeed, (ignoring Bernoulli versus i.i.d. sampling details and the corresponding potential log terms),
> >
> > $$\sqrt{\frac{\mu d \log(d)}{p}}=\sqrt{\frac{\mu d^3 \log(d)}{pd^2}}\sim \sqrt{\frac{\mu d^3 \log(d)}{n}}, $$
> >
> > which seems to imply that one the number of samples required to achieve a given error of $O(\epsilon)$ is $\tilde{O}(d^3/\epsilon^2)$, not $\tilde{O}(d/\epsilon^2)$ (despite the fact that the minimum requirement for these bounds to begin taking effect is still $\tilde{O}(d)$. This would actually be in line with the very early works of Emmanuel Candes.
> >
> > If I didn't make a mistake there, that should **partially invalidate your Remark 7 (depending on interpretation of the dependence on $\sigma$)**, since the bounds are not actually better (in terms of dependence on $d$, where the dependence comes from the last term of equation (21)) than treating each entry independently unless the noise variance $\sigma^2$ is $o(\frac{1}{d})$ which definitely restricts the situation substantially. Note that this doesn’t invalidate either the results in Chen et al or your own results as the case where $\sigma=0$ or $\sigma$ is extremely small ( i.e. $o(\frac{1}{d})$) remains novel, and **even $\sigma=0$ is an interesting case!**
> >
> > *Still, if I am right about this, the writing of the relevant parts of the paper should be updated to include those significant limitations.*
> >
> > Thanks again for all your responsiveness and the nice conversations. I am looking forward to your answer and/or final revision.
> >
> >
> >
> > [edited: fixed typo in rate/condition on $\sigma$ from  $o(\frac{1}{n})$ to  $o(\frac{1}{d})$ ]

---

> > > ### Author Response · Authors · 2022-11-18
> > > **Response to another important question**
> > >
> > >
> > > Thanks for thinking about Chen's results deeply. We believe that the reviewer's initial intuition about the error bound is true i.e. the error does decay as $\sigma \sqrt{\frac{d\log d}{n}}$  where $n$ is the expected number of samples.  We have explained why in details below:
> > >
> > > Note that Chen's final result about the estimate of the low rank matrix says the following (see eq. (5) in the paper):
> > >
> > > $ \|\|\widehat{\mathbf{P}}-\mathbf{P}\|\|_{\infty} \le$
> > >
> > > $O\Big(\frac{\sigma}{\min_i\lambda_{i}}\cdot \sqrt{\frac{\mu d\log d}{p}} \|\|\mathbf{P}\|\|_{\infty}\Big)$
> > >
> > > We believe that the reviewer has missed the factor of $\|\|\mathbf{P}\|\|_{\infty}$
> > >
> > > and $\min_i\lambda_{i}$  in their analysis.
> > >
> > > Note that because of the incoherence assumptions (i.e. the SVD decomposition  $\mathbf{P}=\mathbf{\bar{U}}\mathbf{\Sigma}\mathbf{\bar{V}}^{\mathsf{T}}$
> > >
> > > satisfy  $\|\|\mathbf{\bar{U}}\|\|_{2,\infty}\le \sqrt{\mu r/d}$
> > >
> > > and  $\|\|\mathbf{\bar{V}}\|\|_{2,\infty}\le \sqrt{\mu r/d}$)
> > >
> > > and the fact that the condition number $\kappa=O(1)$ (conditions stated in Lemma 1), we will have
> > >
> > > $||\mathbf{P}||_{\infty} \le $
> > >
> > > $\max_i \lambda_i  ||\bar{\mathbf{U}}||_{2,\infty}$
> > >
> > > $\times ||\bar{\mathbf{V}}||_{2,\infty}$
> > >
> > > $= \max_i \lambda_i  \mu r/d$
> > >
> > > $\implies |\|\mathbf{P}\||_{\infty}/\min_i \lambda_i = \kappa \cdot \mu r/d = O\Big(\mu r/d\Big) $
> > >
> > > (using the fact that $\kappa = O(1)$.
> > >
> > > Substituting the expression of $|\|\mathbf{P}\||_{\infty}/\min_i \lambda_i$ in the error bound, we do get that
> > >
> > > $ |\|\widehat{\mathbf{P}}-\mathbf{P}\||_{\infty} \le O\Big(\sigma\mu r\cdot \sqrt{\frac{\mu d\log d}{pd^2}}\Big)$
> > >
> > > implying that approximately, we will have
> > >
> > > $ |\|\widehat{\mathbf{P}}-\mathbf{P}\||_{\infty} \le O\Big(\sigma\mu r\cdot \sqrt{\frac{\mu d\log d}{n}}\Big)$
> > >
> > > where $n$ is the expected number of samples. Note than $\mu$ and $r$ are usually considered to be constants in matrix completion and so we get the desired rate. We hope this clarifies the question raised by the reviewer.
> > > Please note that a version of this analysis was used in the proof of Lemma 5.

---

> > > > ### Comment · Reviewer_cVaQ · 2022-11-19
> > > > **Thanks**
> > > >
> > > > Dear Authors,
> > > >
> > > > Thanks for the patient explanation. That was a bit of a silly question, I think it has been too long since I last studied that paper more carefully. I indeed misinterpreted the factors of $\frac{\|P\|_\infty}{\lambda_{\min}}$, somehow my I distractedly registered $\frac{\sigma}{\lambda_\in}$ as the conditioning number (temporarily forgetting about the $\sigma$ being the variance).
> > > >
> > > > Thanks for your patience on this one!

---

> > ### Comment · Reviewer_cVaQ · 2022-11-19
> > **Re rebuttals**
> >
> > Dear Authors,
> >
> >
> > Thanks for all the answers and the revisions during the several rounds of reviews. **I think the paper has greatly improved and all the issues have finally been resolved.**
> >
> >
> > **I am happy to recommend this updated version of the manuscript for acceptance and will raise my score to 8.**
> >
> >
> >
> >
> >
> > Congratulations again on the nice paper !

---

### Decision · Program_Chairs · 2023-01-20

**Decision:**

Accept: poster

**Justification For Why Not Higher Score:**

Relatively niche interest. Paper technical but not particularly novel.

**Justification For Why Not Lower Score:**

Well above acceptance threshold.

**Metareview: Summary, Strengths And Weaknesses:**

The paper studies online low-rank matrix completion. In the general (low-rank) case they give a regret bound of O(T^{2/3}) (ignoring polylog factors) and in the special case of rank 1, they give O(T^{1/2}) regret (again ignoring poly log factors). The paper is technical and has required significant back and forth between reviewers and authors, but the reviewers are now confident of correctness (to the extent possible in this kind of review process).

**Note From Pc:**

if the above contains the word "oral" or "spotlight" please see: "oral" presentation means -> notable-top-5% and "spotlight" means -> notable-top-25%. As stated in our emails, we are disassociating presentation type from AC recommendations